**Resource**

# Left ventricular myocardial molecular profile of human diabetic ischaemic cardiomyopathy

Benjamin Hunter [1,2,3], Yunwei Zhang [2,4,5], Dylan Harney [2,3], Holly McEwen [6], Yen Chin Koay [1,2,3], Michael Pan [7,8], Cassandra Malecki [1,2,3,9], Jasmine Khor[3], Robert D Hume [1,2,3,9], Giovanni Guglielmi[3,10,11], Alicia Walker[4,5], Shashwati Dutta[4,5], Vijay Rajagopal [8,10,12,13], Anthony Don [2,3], Mark Larance [2,3], John F O'Sullivan [1,2,3,9,14,15,16,17], Jean Y H Yang [2,4,5,17] & Sean Lal [1,2,3,9,14,15,17] ✉

## Abstract

**Ischaemic cardiomyopathy is the most common cause of heart failure and often coexists with diabetes mellitus, which worsens patient symptom burden and outcomes. Yet, their combined effects are seldom investigated and are poorly understood. To uncover the influencing molecular signature defining ischaemic cardiomyopathy with diabetes, we performed multi-omic analyses of ischaemic and non-ischaemic cardiomyopathy with and without diabetes against healthy age-matched donors. Tissue was sourced from pre-mortem human left ventricular myocardium. Fatty acid transport and oxidation proteins were most downregulated in ischaemic cardiomyopathy with diabetes relative to donors. However, the downregulation of acylcarnitines, perilipin, and ketone body, amino acid, and glucose metabolising proteins indicated lipid metabolism may not be entirely impaired. Oxidative phosphorylation, oxidative stress, myofibrosis, and cardiomyocyte cytoarchitecture also appeared exacerbated principally in ischaemic cardiomyopathy with diabetes. These findings indicate that diabetes confounds the pathological phenotype in heart failure, and the need for a paradigm shift regarding lipid metabolism.**

**Keywords** Human Myocardium; Ischaemic Cardiomyopathy; Diabetes; Multi-omics; Confocal Microscopy
**Subject Categories** Cardiovascular System; Chromatin, Transcription & Genomics; Proteomics

## Introduction

Heart failure (HF) is a global epidemic and leading cause of death affecting 1–2% adults in developed countries, with the most common causes being ischaemic cardiomyopathy (ICM) and non-ischaemic (dilated) cardiomyopathy (NICM) (Groenewegen et al, 2020; Khan et al, 2020; McKenna et al, 2017). ICM is an acquired cardiomyopathy typically due to coronary artery disease (CAD) which primarily affects the left ventricle (LV) (Buja and Vander Heide, 2016; Elosua et al, 2014). NICM can involve both ventricles and is considered to arise from a mix of acquired and genetic aetiologies (Herman et al, 2012; McNally and Mestroni, 2017; Weintraub et al, 2017).

ICM and NICM both express ventricular dilatation and dysfunction and are the most common causes for heart transplantation (Buja and Vander Heide, 2016; McKenna et al, 2017; Stehlik et al, 2011; Weintraub et al, 2017). Yet, ICM transplant recipients have an elevated risk of acute cardiac events and a reduced 10-year survival rate (50%) compared to NICM (84%), often due to underlying risk factors promoting CAD progression, including hypertension and metabolic syndrome (Guddeti et al, 2014). These risk factors also contribute to the development of type 2 diabetes mellitus (DM), which is often concomitant with ICM in HF patients (Bell and Goncalves, 2019; Einarson et al, 2018; Perrone-Filardi et al, 2015). DM, in turn, is associated with dysregulation of fatty acid oxidation (FAO) pathways, with fatty acids being a principal energy source for the heart (Flam et al, 2022; Leguisamo et al, 2012; Lopaschuk et al, 2021; Neglia et al, 2007; Nyman et al, 2013). These metabolic processes, in addition to the extracellular matrix (ECM) and contractile complexes of the myocardium, have not been comprehensively investigated in human hearts, with ICM and DM seldom investigated together but frequently studied in

[1]Precision Cardiovascular Laboratory, The University of Sydney, Sydney, NSW, Australia. [2]Charles Perkins Centre, The University of Sydney, Sydney, NSW, Australia. [3]School of Medical Sciences, Faculty of Medicine and Health, The University of Sydney, Sydney, NSW, Australia. [4]School of Mathematics and Statistics, Faculty of Science, The University of Sydney, Sydney, NSW, Australia. [5]Sydney Precision Data Science Centre, Faculty of Science, The University of Sydney, Sydney, NSW, Australia. [6]School of Biomedical Sciences and Pharmacy, The University of Newcastle, Callaghan, NSW, Australia. [7]School of Mathematics and Statistics, The University of Melbourne, Parkville, VIC, Australia. [8]ARC Centre of Excellence for the Mathematical Analysis of Cellular Systems, The University of Melbourne, Parkville, VIC, Australia. [9]Centre for Heart Failure and Diseases of the Aorta, The Baird Institute for Applied Heart and Lung Surgical Research, Sydney, NSW, Australia. [10]Department of Biomedical Engineering, School of Chemical and Biomedical Engineering, Faculty of Engineering and Information Technology, The University of Melbourne, Parkville, VIC, Australia. [11]School of Mathematics, University of Birmingham, Birmingham B15 2TT, UK. [12]Baker Department of Cardiometabolic Health, Faculty of Medicine, Dentistry, and Health Sciences, The University of Melbourne, Parkville, VIC, Australia. [13]Graeme Clark Institute of Biomedical Engineering, The University of Melbourne, Parkville, VIC, Australia. [14]Central Clinical School, Sydney Medical School, Faculty of Medicine and Health, The University of Sydney, Sydney, NSW, Australia. [15]Department of Cardiology, Royal Prince Alfred Hospital, Camperdown, NSW, Australia. [16]Faculty of Medicine, TU Dresden, Dresden, Germany. [17]These authors contributed equally: John F O'Sullivan, Jean Y H Yang, Sean Lal. ✉E-mail: sean.lal@sydney.edu.au

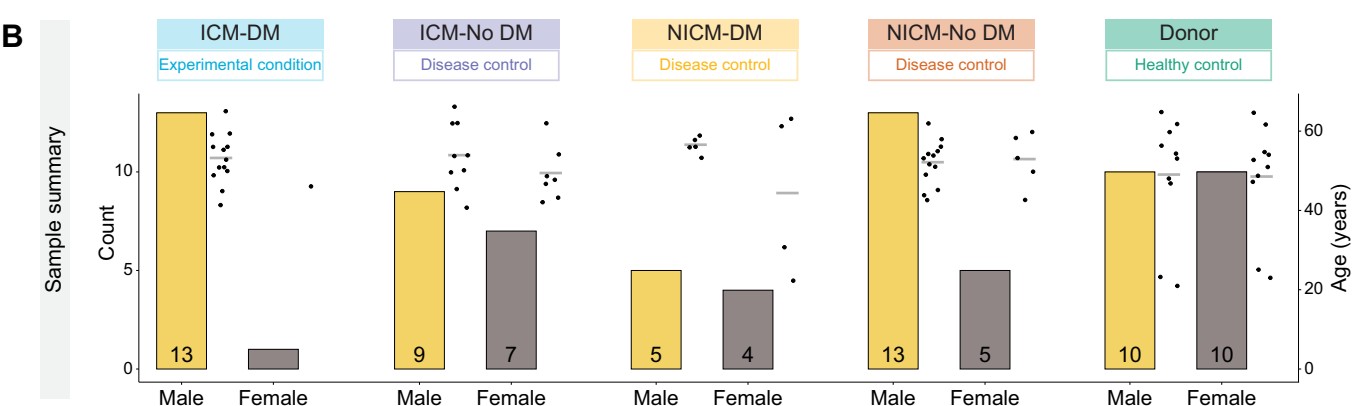

**Figure 1. Overview of experimental design, workflow summary, and sample summary.**

(A) Workflow overview. Created with BioRender.com. (B) Sample summary displaying sex and age of conditions.

isolation (Jia et al, 2018; Paolillo et al, 2019). Given that DM exacerbates HF symptoms and increases mortality in HF (Cubbon et al, 2013; Ebong et al, 2013), there is a need to understand the impact of DM on HF.

Therefore, the aim of this study was to perform an extensive multi-omic analyses on cryopreserved pre-mortem human left ventricular myocardium to uncover the influence of DM in end-stage HF and to delineate the molecular characteristics of ICM with DM (ICM-DM), thereby creating a comprehensive molecular resource of DM-related HF.

## Results

Myocardial multi-omic mass spectrometry (MS) differential analyses were performed on all end-stage HF conditions vs healthy age-matched donors (AMD) wherein DM and no DM groups were contrasted against each other (Fig. 1A). A sample summary is depicted in Fig. 1B. Demographic data can be seen in Dataset EV1. Multidimensional scaling (MDS) plots were used to visualise individual and group similarity via a distance matrix for each MS analysis (Fig. EV1A–L).

### Differential molecular analysis of ischaemic cardiomyopathy with diabetes

Analyses of HF conditions with and without diabetes compared against AMD can be found in Appendix Fig. S1 and Datasets EV2–EV7. When focused on individual HF conditions vs AMD, proteomic analysis resulted in 593 downregulated and 654 upregulated proteins in ICM-DM of which, 1098 were only differentially abundant (DA) in the ICM-DM group of ICM, such as regulator of microtubule dynamics (RMDN3), α-actinin 1 (ACTN1), mitochondrial isocitrate dehydrogenase (IDH2), and mitochondrial acetyl-CoA acetyltransferase (ACAT1) (Fig. 2A,D,G; Datasets EV8 and EV11). Cross-comparison of HF conditions vs AMD identified serum amyloid A1 (SAA1) was uniquely and greatly downregulated in ICM-DM. Other highly DA proteins specific to ICM-DM included neogenin (NEO1), numerous NDUF complex I (CI) subunits such as NDUFS3 and muscle aldolase (ALDOA) (Fig. EV2A,D,G; Datasets EV15 and EV18). Proteins specifically DA in DM groups included multiple NDUFs, cysteine protease inhibitor FETUB, acyl-CoA oxidase (ACOX2), mitochondrial acetyl-CoA acyltransferase (ACAA2), and extracellular matrix (ECM)-related cytoskeletal protein talin 1 (TLN1). Some of the most significant ICM-DM proteins were also shared with the other HF groups, including complement factor D (CFD), glutamic-pyruvic transaminase 2 (GPT), and mitochondrial acyl-CoA synthetase (ACSS3).

In the metabolomic analysis, there were 11 downregulated and 11 upregulated metabolites in ICM-DM vs AMD of which 7 were DA in ICM-DM only in the ICM group. Oxaloacetate, acetylcholine, nicotinamide adenine dinucleotide phosphate, glycine,

arginine and cyclic adenosine monophosphate (cAMP), a second messenger for intracellular signalling, were among the metabolites uniquely DA in ICM-DM relative to the other forms of HF (Figs. 2B,E,H, EV2B,E,H, and EV4D,E; Datasets EV9, EV13, EV16, and EV19). Metabolites which were uniquely upregulated in ICM included fructose 1,6-bisphosphate and adenosine diphosphate (ADP). Alanine (downregulated) and isoleucine/leucine amino acids (upregulated) were found to be commonly DA in all HF conditions in the same direction. Cystamine featured the greatest FCs and was upregulated in ICM-DM, ICM-No DM, and NICM-No DM. $NAD^+/NADH$ ratio was upregulated in ICM-DM, ICM-No DM, and NICM-No DM vs AMD. $NADP^+/NADPH$ ratio was downregulated in ICM-DM, ICM-No DM, and NICM-No DM.

Identified HF-specific signature proteins and metabolites can be observed in Fig. EV4A,D and Datasets EV65–71.

Lipidomic analysis identified 20 downregulated lipids in ICM-DM, 15 of which were medium-chain (4), long-chain (10), and very-long-chain (1) acylcarnitines, as well as diacylglyerols (2) and phosphatidylglycerols (2), with no other HF condition presenting FDR significant lipid differential abundance individually (Figs. 2C,F,I, EV2C,F,I, and EV4F; Datasets EV10, EV14, EV17, and EV20).

To allow exploration of these differential analyses, we produced an interactive resource (shiny app, http://shiny.maths.usyd.edu.au/Human-Heart-ALL/).

### Enriched metabolic pathways and gene sets

To investigate implicated pathways, Gene Set Enrichment Analysis (GSEA) protein enrichment analyses via clusterProfiler were performed on the normalised data using targeted Kyoto Encyclopedia of Genes and Genomes (KEGG) and MitoCarta3.0 databases (Figs. 3A and EV3A; Datasets EV21–EV24). Selected enriched mitochondrial/energetics-associated pathways/gene sets were annotated, ranked, and cross-HF group comparisons were performed (Figs. 3C,D,G and EV3C,D).

Lipid metabolism, atherosclerosis, and cholesterol metabolism pathways were positively enriched in HF. Cholesterol metabolism was specifically enriched in ICM-DM, ICM-No DM, and NICM-DM. This enrichment was largely attributed to the upregulation of intramyocardial chylomicron apolipoproteins, including APOA4, APOB, and APOC3, in all HF conditions, with all but one (APOL6) being upregulated in ICM-DM. There were many common negatively enriched pathways/gene sets such as lipid metabolism, carbohydrate metabolism, thermogenesis, and diabetic cardiomyopathy in all HF groups. Additional pathways/gene sets were negatively enriched only in ICM-DM, such as the citrate cycle, mitochondrial RNA, fatty acid, branched-chain amino acid (BCAA), and lysine metabolism. Another notable negatively enriched metabolic pathway was propanoate metabolism, along with the peroxisome gene set, which were shared with NICM.

Oxidative phosphorylation (OXPHOS) and many of its subdivided gene sets were also observed as negatively enriched in

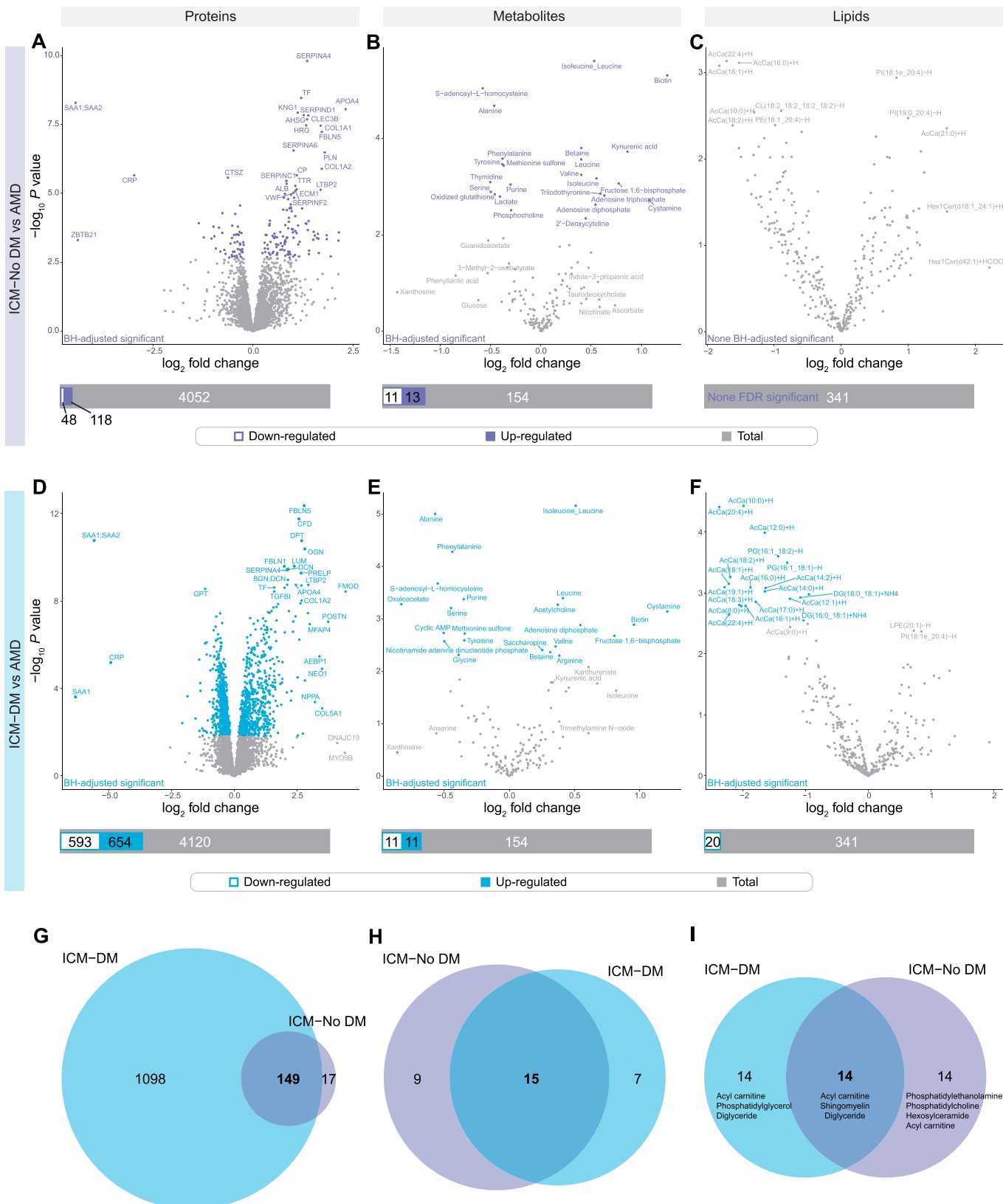

◄ **Figure 2.   Human left ventricular myocardial differential analysis in protein, metabolite, and lipid abundance between ischaemic cardiomyopathy with (ICM-DM) and without diabetes (ICM-No DM) and age-matched donors (AMD).**

(A–F) Differential abundance was determined following Benjamini–Hochberg false discovery rate adjustment (FDR) of $P$ values (FDR < 0.05). Analyses were performed using a moderated $t$ test with the limma package (version 3.56.2) in R (version 4.3.1) following $\log_2$ transformation. ICM-DM $n = 14$, ICM-No DM $n = 16$, AMD $n = 20$ (proteomics and metabolomics) and 19 (lipidomics). Superimposed bar plots summarise the number of significantly downregulated (white-filled bar) and upregulated (colour-filled bar) molecules relative to the total number of molecules analysed (grey bar). (A–C) ICM-No DM vs AMD. (D–F) ICM-DM vs AMD. (G, H) ICM-DM and ICM-No DM vs AMD FDR significantly differentially abundant proteins and metabolites. (I) ICM-DM vs AMD and ICM-No DM vs AMD unadjusted significant ($P$ < 0.05) lipids, which were also > ±2-fold change (FC, in either direction). Lipid classes annotated in order of the number of lipids which were unadjusted significantly.

all HF conditions, however, showed greater enrichment in ICM-DM, wherein a greater number of the contained gene/proteins were FDR significant from the differential analysis (90 OXPHOS genes DA in ICM-DM out of 169; NICM-No DM, 35; NICM-DM, 27). All were downregulated except for the upregulation of HIG1 hypoxia inducible domain family member 1A (HIGD1A). CI subunits were most enriched in ICM-DM (enrichment score = −0.85) followed by NICM-DM (enrichment score = −0.76). Other proton pumping complexes and their subunit gene sets; CIII, CIV, and CV, were similarly negatively enriched in ICM-DM wherein CIII and CIV were co-enriched in NICM-DM and CV was ICM-DM specific. Gene sets mediating electron transport for OXPHOS, such as iron–sulfur cluster biosynthesis (enriched in DM) coenzyme Q metabolism (enriched in ICM), and electron carriers were also negatively enriched in ICM-DM.

The clusterProfiler KEGG enriched pathway analysis on the normalised metabolomic MS data identified a positive enrichment in histidine metabolism and a negative enrichment in thyroid hormone signalling pathways exclusive to ICM-DM vs AMD (Figs. 3B and EV3B; Datasets EV25–EV28).

## Extracellular matrix, cytoskeletal, and sarcomeric enriched gene sets

GSEA Gene Ontology (GO) enrichment analysis was also performed on selected cytoskeletal, sarcomeric, and ECM parent nodes from the Biological Process (BP) and Cellular Component (CC) libraries vs AMD (Figs. 3E,F,H and EV3E,F; Datasets EV29–EV32). Cardiomyocyte contraction and conduction-related nodes were seen to be negatively enriched specific to ICM conditions, whereas regulation of cardiac muscle contraction by calcium ion signalling was unique to ICM-DM. This enrichment was largely attributable to the downregulation of cardiac troponin I (TNNI3), sarcoplasmic reticulum Ca2+ ATPase (ATP2A2), Na+/Ca2+ exchanger (SLC8A1), ryanodine receptor (RYR2), and calsequestrin 2 (CASQ2) as determined from the differential analysis. ICM-DM-specific positive enrichment was observed in many actin/cytoskeletal nodes such as Arp2/3 protein complex, podosome, actin filament, and stress fibre nodes, where cytoplasmic microtubule (strongly influenced by SAA1, TPT1, and MTUS2) was negatively enriched. GO nodes only positively enriched in DM included barbed-end actin filament capping, actin cap, and actomyosin structure organisation and actomyosin nodes wherein ICM-DM showed deviation from NICM-DM and No DM in higher myosin heavy chain (MYH10, MYH11, MYH9, and MYL9) FCs. Multiple ECM nodes were positively co-enriched in all HF conditions, but more proteins were observed to be FDR upregulated in ICM-DM from the differential analysis, such as in elastic fibre assembly and collagen fibril organisation, strongly influenced by collagens (COLs), with unique upregulation in ICM-DM in proteins such as FKBP prolyl isomerase 10 (FKBP10) and elastin microfibril interfacer 1 (EMILIN1). We previously reported an increased elastic fibre assembly and elastogenesis within both rat and human ICM (Hume et al, 2023). Cytoskeletal node, actin filament assembly, was also positively co-enriched in HF.

## Ischaemic cardiomyopathy with diabetes and diabetes-specific targeted analyses

Supplementary GO enrichment analyses via PANTHER from significantly DA proteins vs AMD revealed gene sets which were commonly enriched in HF (Fig. EV4A,B; Datasets EV33 and EV34). Analyses also identified ICM-DM-specific impaired mitochondrial metabolism due to the downregulation of alpha-ketoglutarate, pyruvate and acyl-CoA dehydrogenases (ACAD8, ACAD9, ACADVL, GCDH), acetyl-CoA C-acyltransferase (HADHA, HADHB, ACAA1) and 3-methylcrotonyl-CoA carboxylase activity (Fig. EV4A,C; Dataset EV35). Positively enriched ICM-DM-specific gene sets included cadherin binding and galactose metabolic process (GALM, B4GALT1, GALE).

A diabetes-specific targeted analysis showed negatively enriched Z-disc and contractile gene sets enriched due to the downregulation of CASQ2, peripherin (PRPH), nebulette (NEBL), titin-cap (TCAP), and LIM domain-binding proteins LDB3, PDLIM4, and PDLIM5 (Appendix Fig. S2A,B; Dataset EV36). OXPHOS subunits of C1 (NDUFA5, NDUFB10, and NDUFV2), CIV, cytochrome C oxidase subunit 7C (COX7C), coenzyme Q9 (COQ9), and COQ10, were also downregulated specifically in DM.

As the ICM-DM group had a higher BMI (30.8 ± 5.1) than the other conditions, a BMI-average molecule abundance Pearson correlation test was also performed to identify a potentially significant contributory influence to the results from a high BMI/obesity, whereby, no significance was identified (Appendix Fig. S2C–E; Datasets EV37–EV39).

## Mitochondria and oxidative phosphorylation simulated modelling

To study the potential physiological consequences of reduced CI activity in ICM-DM (Fig. 4A), we employed a biophysical model of OXPHOS (Fig. 4B–D). The model was validated using rat cardiomyocyte OXPHOS data due to its availability over human data. Given the metabolic differences between rat and human cardiac myocytes (Milani-Nejad and Janssen, 2014), the model could not make quantitative predictions but was used to infer trends in physiological variables related to decreased CI activity.

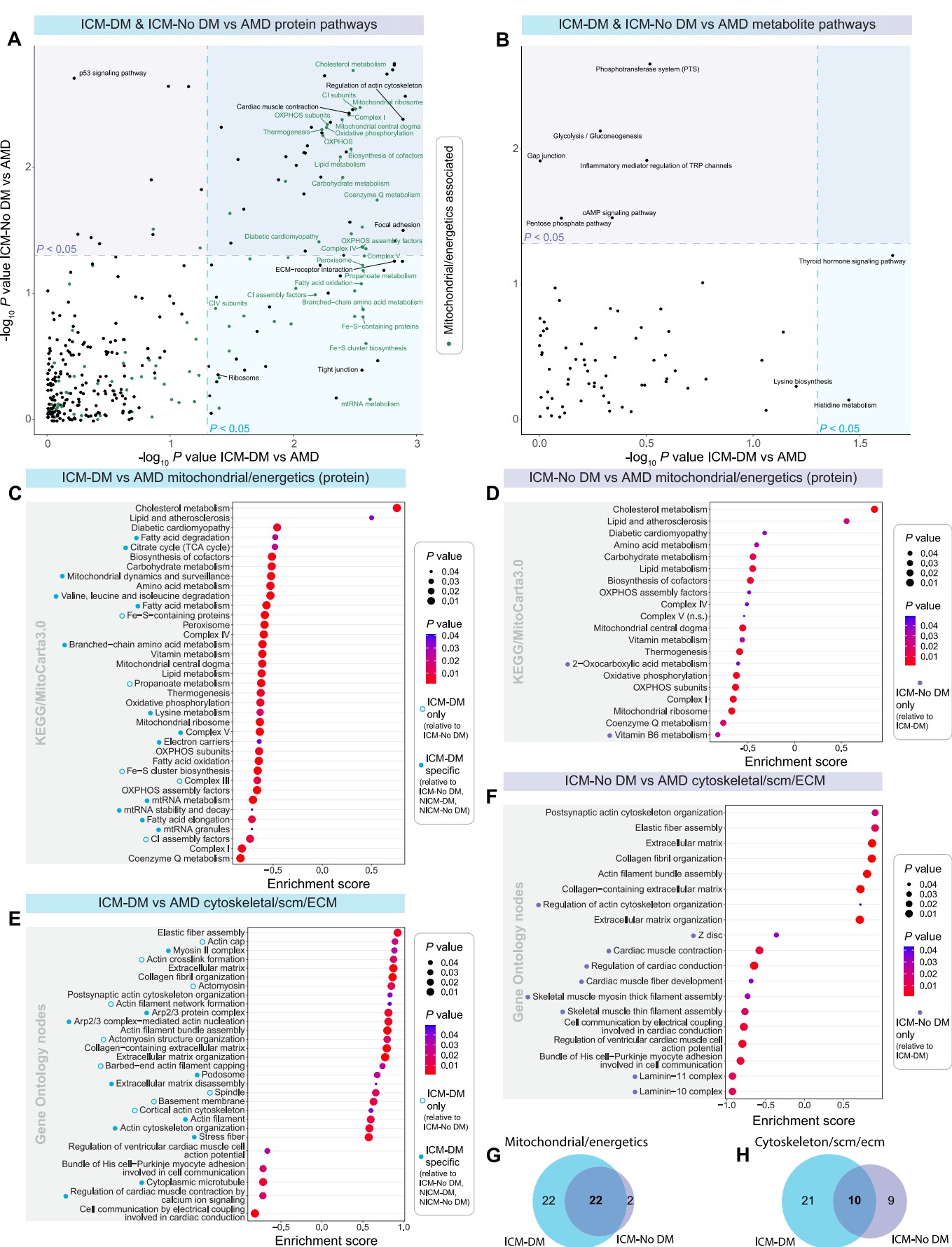

**Figure 3. Human ischaemic cardiomyopathy with (ICM-DM) and without diabetes (ICM-No DM) vs healthy age-matched donor (AMD) protein and metabolite pathway analyses.**

(A, B) Scatter plots of $-\log_{10} P$ value enriched KEGG and MitoCarta3.0 protein pathways/gene sets of ICM-DM vs AMD and ICM-No DM vs AMD where mitochondrial and energetics related pathways are coloured in green and significant ($P < 0.05$). Pathways in overlapping coloured regions were significant in both ICM-DM and ICM-No DM vs AMD. (A) Enriched pathways from the proteomic mass spectrometry (MS) analysis. (B) Enriched pathways from the metabolomic MS analysis. (C–F) Gene Set Enrichment Analysis (GSEA) enrichment bubble plots of top-ranked significantly enriched ($P < 0.05$) KEGG/MitoCarta3.0 pathways/gene sets and Gene Ontology nodes, respectively. (C, D) From (A). (E, F) Cytoskeletal, sarcomeric (scm), and extracellular matrix (ECM) enriched Gene Ontology Biological Process and Cellular Component nodes from selected parent nodes. (G) Mitochondrial/energetics-associated pathways from (A). (H) All enriched Gene Ontology nodes which produced (E, F). Gene Set Enrichment Analysis (GSEA) analyses for (A–F) were performed on normalised and transformed proteomic and metabolomic datasets using clusterProfiler (version 4.8.1).

The simulations showed that decreased CI activity alone has little effect on the function of OXPHOS, which is consistent with results in prior theoretical studies (Wu et al, 2007) and experimental studies in animals (Lucas and Szweda, 1999; Valsecchi et al, 2012). Figure 4E depicts an intra-mitochondrial schematic representing all DA proteins in ICM-DM vs AMD directly involved in OXPHOS.

Leveraging the opportunity of analysing the pre-mortem human myocardium in healthy and end-stage HF conditions, we sought the protein with which expression per individual best correlated with the combined average expression of the mitochondrial proteome (MitoCarta3.0) as a candidate for predicting mitochondrial volume in ICM-DM, HF, or any condition. Succinate-CoA ligase SUCLG1 was the strongest predictor of ICM-DM mitochondrial volume ($\rho = 0.958$), while mitochondrial creatine kinase (CKMT2) ($\rho = 0.948$) and fumarate hydratase (FH) ($\rho = 0.932$) were best for HF. Enoyl-CoA hydratase ECHS1 ($\rho = 0.889$) was most suitable across all conditions (Fig. 4F; Datasets EV40–EV42). These were alongside previously referenced proteins as predictors of mitochondrial volume/OXPHOS; citrate synthase (CS) (Kumar et al, 2019; Larsen et al, 2012), heatshock protein HSPD1 (Morgenstern et al, 2021), and LYR motif-containing protein 1 (LYRM1) (McLaughlin et al, 2020). Investigation of proteins regulating mitochondrial morphology identified the downregulation of elongation factors G1 (GFM1) and Tu (TUFM), and fission protein dynamin 1-like (DNM1L) in ICM-DM (Fig. 4G).

## RNA sequencing analysis, and RNA–protein correlation, and co-regulation

To investigate whether significantly DA proteins may have been regulated at the transcription or post-translational level, or to identify differentially expressed noncoding sequences which could regulate protein expression, RNA sequencing (RNA-seq) was performed on selected ICM-DM ($n = 7$), ICM-No DM ($n = 7$) and AMD ($n = 7$) left ventricular myocardial tissues (Dataset EV43). It was found that 120 transcripts were downregulated and 1307 were upregulated in ICM-DM vs AMD. Of these, 334 were differentially expressed in both ICM-DM and ICM-No DM. The 1093 transcripts which were only differentially expressed in ICM-DM showed significant enrichment in the extracellular matrix GO CC node (FDR = $7.80 \times 10^{-17}$) and the ECM-receptor interaction KEGG pathway (FDR = 0.0233) via STRING analysis (Fig. EV5A–C; Datasets EV44 and EV46).

Paired RNA and protein $\log_2$ FC of ICM groups were plotted to compare FC similarities compared to AMD (Figs. 5A,B and EV5F). An RNA–protein Pearson correlation analysis was performed in each condition wherein 179 RNA transcripts and proteins were

correlated only in ICM-DM, and nestin (NES), a biomarker for mitosis, was the only gene symbol co-correlated in all conditions (Fig. 5C; Datasets EV48–EV50). NES protein was upregulated in all HF conditions, with the greatest FCs in DM, and was RNA–protein co-regulated in ICM-DM.

It was found that 74 gene symbols were RNA–protein co-regulated in ICM-DM vs AMD in the same direction, 66 of which were only observed in ICM, and all ICM-No DM co-regulated gene symbols were shared by ICM-DM (Fig. 5D–F; Dataset EV52). A GO chord plot of selected significantly enriched CC nodes was formed to summarise the RNA and protein co-regulated only in ICM-DM showing that 49 of the 66 gene symbols were contained within extracellular, cytoskeletal, and focal adhesion gene sets (Fig. 5G). Of these, 9 gene symbols were also RNA–protein correlated such as natriuretic peptide A (NPPA), adipocyte enhancer-binding protein 1 (AEBP1), fascin (FSCN1), and COL1A2 (Fig. 5H–K).

A correlation analysis of all groups combined identified proteins likely constitutively expressed at the transcription level irrespective of HF (Fig. EV5G).

Analysed extracellular matrix secretome proteins and RNA have been isolated in Datasets EV72 and EV73 (Uhlén et al, 2019).

## Ischaemic cardiomyopathy with and without diabetes comparison

Further targeted analyses via PANTHER using the differential analysis results investigated the comparison between ICM-DM and ICM-No DM. Commonly downregulated proteins vs AMD revealed negative enrichment of the sodium:potassium-exchanging ATPase complex gene set unique to ICM (Fig. EV6A,B; Dataset EV34). ICM common DA transcripts produced positive enrichment of ECM and cell adhesion gene sets (Fig. EV6E,F; Dataset EV47).

Proteins which were ICM-DM characteristic and divergent from ICM-No DM included upregulation of collagen-containing extracellular matrix-related proteins, dermatopontin (DPT), transforming growth factor β induced (TGFBI), COL18A1, and annexin 4 (ANXA4), and downregulation of COP9 signalosome subunit 8 (COPS8), NDUFAF2, and perilipin 4 (PLIN4) (Fig. EV6C,D). An RNA–protein correlation analysis also identified ICM-DM-specific enriched pathways/gene sets (Fig. EV5H–J; Datasets EV53 and EV54).

Numerous cardiac pathology-associated coding (Fig. EV5D,E), and noncoding sequences such as long noncoding RNAs (lncRNA), microRNAs (miRNA), and antisense RNAs, which are known to regulate gene expression, were also identified to be ICM-DM-specific and divergent (Fig. EV6G–I).

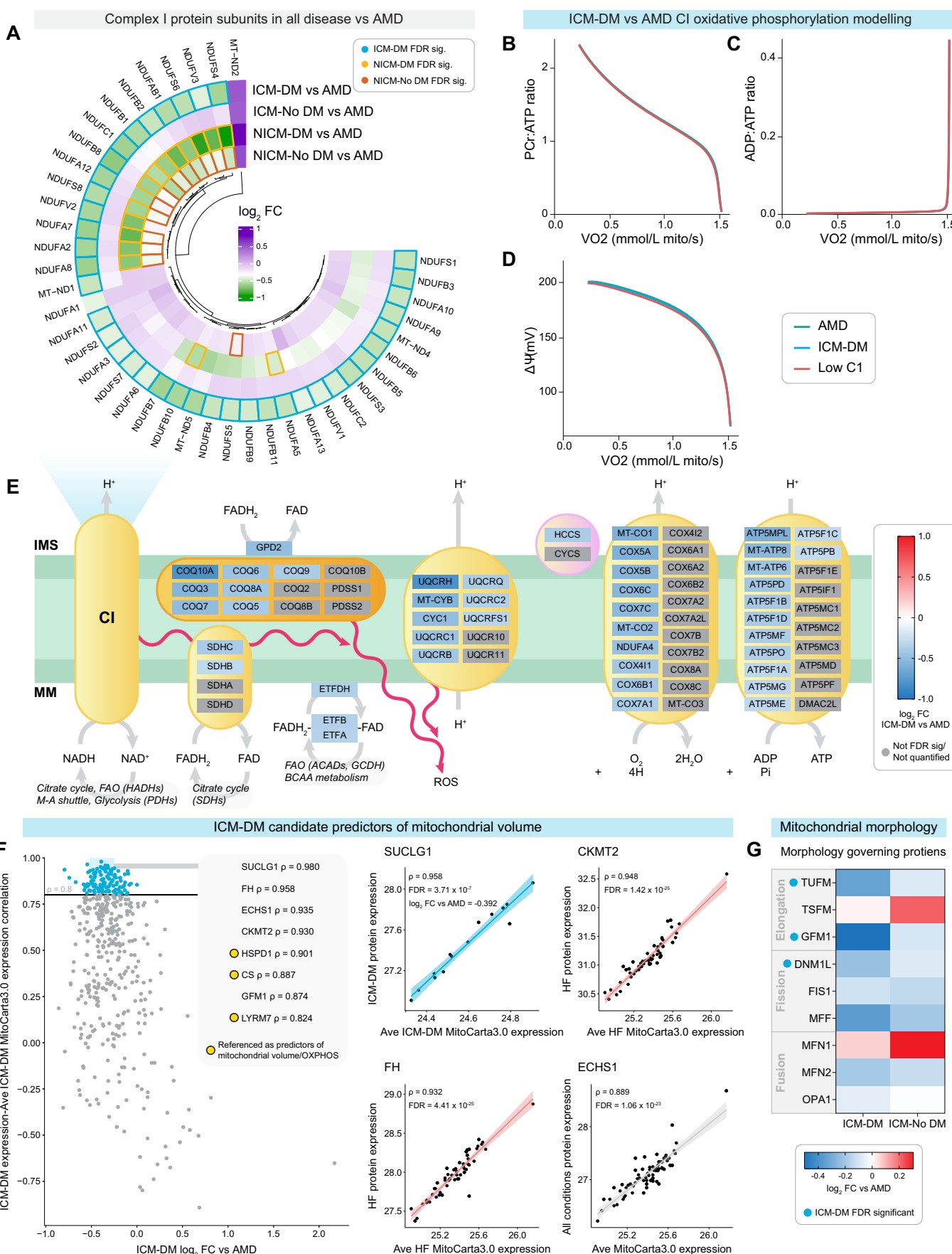

**Figure 4. Ischaemic cardiomyopathy with diabetes (ICM-DM) influence on oxidative phosphorylation, mitochondrial composition, and mitochondrial morphology.**

(A) Quantified electron transport chain complex I (CI) subunits from proteomic mass spectrometry (MS) showing the $\log_2$ fold change (FC) of all heart failure (HF) conditions; ICM-DM, ICM without diabetes (ICM-No DM), non-ischaemic cardiomyopathy with diabetes (NICM-DM), and NICM without diabetes (NICM-No DM), vs age-matched donors (AMD). Significance was determined after Benjamini–Hochberg false discovery rate adjustment (FDR) of $P$ values (FDR < 0.05). (B–D) Simulations of the computational model of oxidative phosphorylation. The groups correspond to AMD myocytes (green; baseline CI activity), ICM-DM myocytes (blue; complex I activity −29.4%, based on proteomic MS) and myocytes with even lower CI expression (red; CI activity −40%). (D) The mitochondrial membrane potential (ΔΨ) are plotted against the oxygen consumption rate $VO_2$. (E) Schematic of oxidative phosphorylation depicting FDR significant proteins, where colour represents $\log_2$ FC in ICM-DM vs AMD. Proteins not FDR significant or not quantified/analysed represented in grey. CI to CV; left to right. IMS intermembrane space, MM mitochondrial matrix, FAO fatty acid oxidation, ROS reactive oxygen species. (F) $\log_2$ FC of ICM-DM mitochondrial proteome vs AMD and correlation of ICM-DM individual mitochondrial protein expression to the average relative expression of the ICM-DM mitochondrial proteome across each patient. Only proteins quantified in all samples were analysed. (G) Quantified proteins influencing mitochondrial morphology.

## Sex influence in multi-omic MS analyses

As all ICM-DM patients in the RNA-seq cohort and all but one in the multi-omic analyses were male, donor male vs female comparisons were performed to assess potential sex-based confounding. No significant differential abundance or separation by sex was observed (Fig. EV7A–F; Datasets EV55–EV58). ECM and fibrosis regulator AEBP1 was upregulated in all HF conditions, with the greatest FC and RNA–protein co-regulation in ICM-DM. Although AEBP1's expression appeared to correlate with male predominance and FC increases in HF vs AMD, it was not significant (Fig. EV7G).

## Histochemistry

Histochemistry was performed to qualitatively observe the myocardial changes in ICM-DM described by proteomic MS. Diffuse interstitial and perivascular collagen fibril staining was observed in ICM vs comparable AMD myocardium, representing increased fibrosis (Fig. 6A–C). Macroscopic infarct/replacement fibrosis was avoided in all analyses (Appendix Fig. S3A,B). Key proteins of interest were labelled and imaged via immunofluorescent confocal microscopy, with cardiomyocytes longitudinally oriented, based on downregulated proteomic MS results and their biological/cited significance. ATP2A2/SERCA2 showed disruption/disorganisation within ICM-DM cardiomyocytes, particularly along the Z-discs, vs AMD, with TNNI3 co-localising along the I-band (Figs. 6D–F,M,N and EV8M; Appendix Fig. S3C,D). Increased interstitial space was noted between the sarcolemma of ICM-DM cardiomyocytes, with broader and disorganised staining of intramembrane glycoproteins indicating ECM linkage dysfunction. Disruption was also indicated in OXPHOS CI subunit NDUFS3 and mitochondrial elongation factor GFM1 proteins at the I-band (Figs. 6G–L,O,P and EV8M; Appendix Fig. S3E,F). Mitochondria were also labelled with Mito-ID where bright aggregates were attributed by the compounding mitochondrial OXPHOS-associated flavin autofluorescence with the highest concentrations in the perinuclear region signifying a notable reduction of both in ICM-DM. Majority of the flavin at this emission wavelength (500–550 nm) was assumed to be in the form of free flavin adenine dinucleotide (FAD), followed by flavin-bound flavoproteins (ETFs) with the least attributed to acyl-CoA dehydrogenases (Fig. EV8A–D,K) (Chorvat et al, 2004; Chorvat et al, 2005; Kunz and Gellerich, 1993; Lu et al, 2014; Romashko et al, 1998; Wust et al, 2015). ICM-DM T-tubules between mitochondria presented disorganisation at Z-disc locations. Glycolytic muscle aldolase

(ALDOA), which binds to F-actin of the sarcomere and regulates cardiac hypertrophy (Clarke and Morton, 1976; Li et al, 2018b), was regularly distributed across the Z-disc in AMD but appeared dislocated in ICM-DM (Fig. EV8E–J; Appendix Fig. S3G,H).

## Discussion

We have created a resource on the differential abundance of proteins, metabolites, lipids, and RNA in human left ventricular samples from end-stage ICM-DM compared to AMD and ICM-No DM, and in reference to NICM with and without DM.

### The interactions between fatty acid oxidation and lipolysis

Our results identified the upregulation of circulating chylomicron apolipoproteins (APOs) in ICM-DM, suggesting that diabetes may drive reduced FAO. APOs are positively associated with high blood pressure and atherosclerosis and have been used as an indicator of metabolic syndrome and cardiac stress (Agarwala et al, 2017; Clarke et al, 2023; Gofman et al, 1950; Hunter et al, 2019). The downregulation of components associated with the peroxisome proliferator-activated receptor (PPAR)-α pathway, of long-chain and very-long-chain fatty acid transporters (SLC27A1, CD36), cytosolic fatty acid binding proteins, acyl-CoA synthetases, and proteins involved in FAO, indicates a reduced utilisation of FA in ICM-DM. However, there was an upregulation of albumin (ALB) and transmembrane medium-chain fatty acid transport (LRP1) in addition to a downregulation of lipid droplet-surrounding perilipin (PLIN4) in ICM-DM, which may suggest an increased bioavailability of free fatty acids (FFAs) in the myocardium. Short-chain FAO may be unhindered or even upregulated, as identified in a TAC-induced HF rat model, (Carley et al, 2021) as they do not require transmembrane or intracellular facilitatory transport.

It has been previously identified that ATP synthesis, of which FAO normally accounts for ~50–70% within the healthy myocardium, undergoes a metabolic shift to carbohydrate and ketone reliance in HF (Karwi et al, 2018; Lopaschuk et al, 2021; Stanley et al, 2005). There is evidence that the heart prioritises glucose metabolism over FAO as a source of acetyl-CoA in the presence of oxidative stress or metabolic dysfunction (Bogh et al, 2020; Dyck et al, 2004; Schroeder et al, 2012). Yet, paired with our proteomic results, medium-chain, long-chain, and very-long-chain acylcarnitines were downregulated in HF vs AMD, particularly in ICM-DM, as well as diacylglycerol, an intermediate to triacylglycerol synthesis

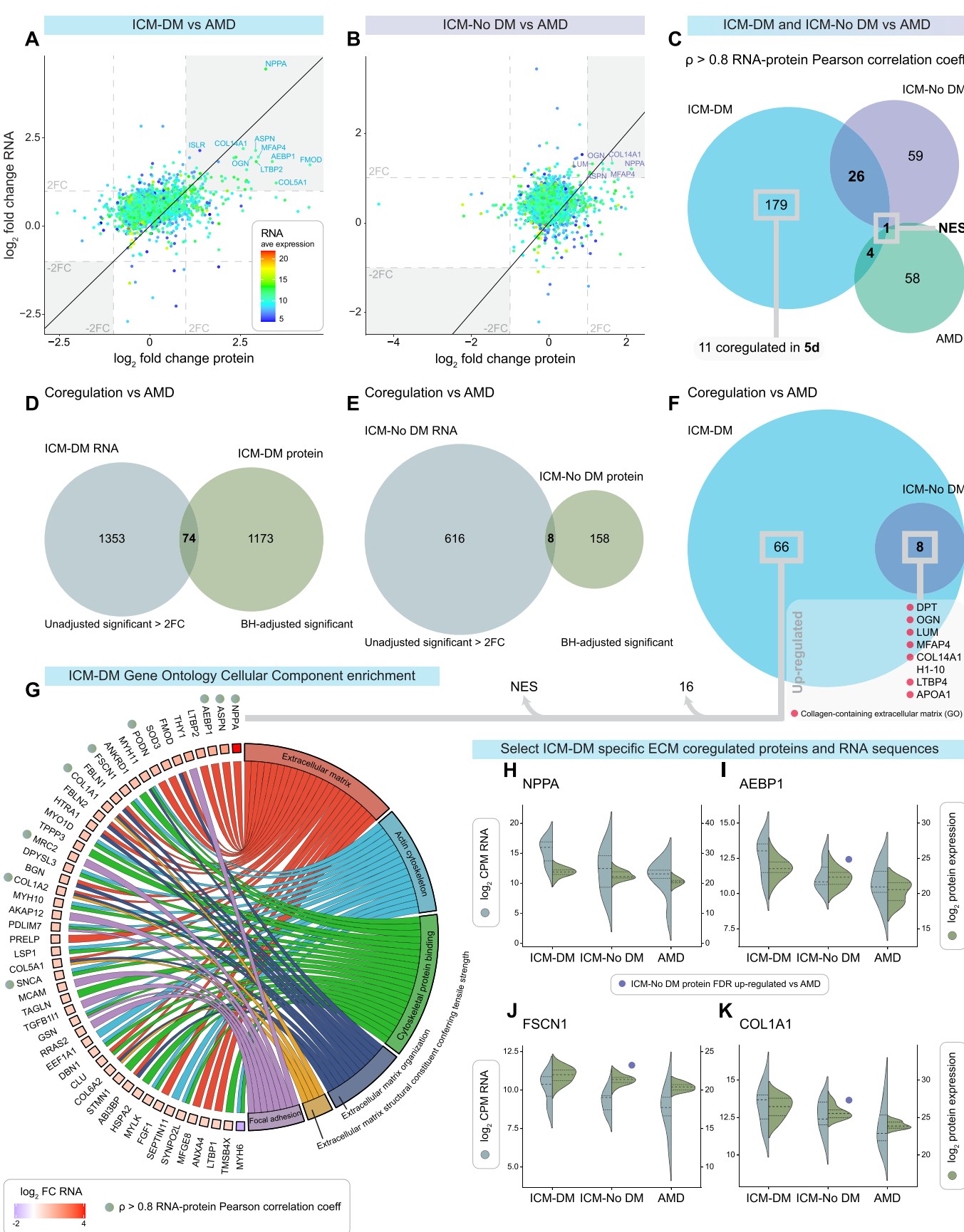

**Figure 5.   Ischaemic cardiomyopathy with (ICM-DM) and without diabetes (ICM-No DM) RNA–protein correlation and co-regulation vs age-matched donors (AMD).**

(A) RNA and protein $\log_2$ fold change (FC) plot of ICM-DM vs AMD. ICM-DM average RNA expression represented on a colour scale. (B) RNA and protein $\log_2$ FC plot of ICM-No DM vs AMD. (C) Gene symbols which had an RNA–protein Pearson correlation coefficient of $\rho > 0.8$ in ICM-DM, ICM-No DM and AMD. (D) ICM-DM vs AMD significantly differentially expressed RNA ($P < 0.05$ and $> \pm 2$-FC) and protein (FDR $< 0.05$) gene symbols. (E) ICM-No DM vs AMD significant RNA and protein gene symbols. (F) From (D, E), including list of gene symbols RNA–protein co-regulated in both ICM-DM and ICM-No DM vs AMD. (G) Gene Ontology Cellular Component enrichment chord plot of extracellular, cytoskeletal and focal adhesion related gene symbols from (F) which were co-regulated only in ICM-DM vs AMD with RNA $\log_2$ FC represented. RNA–protein Pearson correlation coefficient of $\rho > 0.8$ indicated. (H–K) Selected gene symbols of interest from (G) showing $\log_2$ transformed RNA and protein expression. Purple circles highlight the gene symbols which were also ICM-No DM vs AMD significantly differentially expressed (FDR $< 0.05$) in the proteomic analysis. ICM-DM $n = 14$ (proteomics) and 7 (RNA-seq), ICM-No DM $n = 16$ (proteomics) and 7 (RNA-seq), AMD $n = 20$ (proteomics) and 7 (RNA-seq).

and lipid droplet formation, suggesting an increased fatty acid metabolism dependence or efflux/loss of fatty acids into the circulation in ICM-DM relative to other HF; used or lost. This concept of lipid efflux, supported by positive myocardial APOA1 RNA–protein co-regulation in ICM, to our knowledge, has not been described in the myocardium but been observed in chick skeletal muscle in response to lipid overload (Shackelford and Lebherz, 1983; Tarugi et al, 1989).

Increased plasma acylcarnitines have been attributed as a characteristic of HF, including ICM-DM, and to identify high-risk patients (Ruiz et al, 2017; Wilshaw et al, 2022; Yoshihisa et al, 2017; Zhao et al, 2020). Our results suggest downregulation of FAO in HF as well as an association with reduced myocardial tissue acylcarntines, but there is a lack of consensus regarding whether FAO is down or upregulated in HF and ICM-DM (Bedi et al, 2016; Fillmore et al, 2014; Karwi et al, 2018; Martín et al, 2000; Stride et al, 2013). There was a downregulation of PLIN4 in ICM-DM only, indicating the initiation of intracellular lipolysis and the release of FFAs for FAO. It has previously been shown that inactivation of PLIN4 results in the downregulation of PLIN5 RNA and protein expression in the heart compared to other organs (Chen et al, 2013; Kien et al, 2022). Reduced PLIN5 has been attributed to exaggeration of the HF phenotype, increased FAO and oxidative stress (Zhou et al, 2019), and has been considered a therapeutic target to prevent myocardial lipid accumulation (Cui et al, 2022).

Myocardial lipogenesis and the resulting formation of lipid droplets have been a well-documented yet poorly understood feature of HF, ischaemia, insulin resistance and diabetes, obesity and metabolic syndrome (Goldberg et al, 2018; Goldenberg et al, 2019; Krahmer et al, 2013; Lee et al, 2013; McGavock et al, 2007; Nyman et al, 2013; Sletten et al, 2018; Wende and Abel, 2010). This condition has been referred to as cardiac steatosis. We identified an overall reduced myocardial lipid content, and to our knowledge, this process has not been investigated in human end-stage ICM-DM.

### Indications of impaired glucose metabolism

The end-stage ICM-DM heart showed signs of promoting glycolysis via intracellular glucose availability, as anticipated in HF and ischaemia due to pathological hypertrophy stimuli and oxidative stress (Dyck et al, 2004; Fillmore et al, 2014; Ritterhoff and Tian, 2023; Shao and Tian, 2016), by the upregulation of glucose trapping hexokinase (HK1, the first rate-limiting step of glycolysis), which was also observed in NICM in agreement with a recent study (Flam et al, 2022).

Diabetes and insulin resistance have been shown to present with hyperglycaemia and impaired insulin-sensitive glucose transporter (SLC2A4/GLUT4) translocation to the plasma membrane and a reduced expression in the rat heart (Leguisamo et al, 2012). Our results also show the downregulation of SLC2A4 in addition to an upregulation of basal glucose transporters (SLC2A1/GLUT1) in ICM-DM with neither DA in other forms of HF. This is in concordance with an early study in dogs (Lei et al, 2004), however, an upregulation of SLC2A1 was identified in rats with coronary ligation-induced HF (Rosenblatt-Velin et al, 2001). As failing and ischaemic hearts have a greater dependence on glycolysis, impaired expression and translocation of SLC2A4 to the sarcolemma due to diabetes could result in an exacerbated reduction in systolic function (Tian and Abel, 2001).

### Reduced expression of ketone and BCAA metabolism proteins

Ketone body, particularly 3-hydroxybutyrate, uptake and metabolism are increased in HF, likely due to FAO being reduced, and has also been proposed as treatment for HF (Bedi et al, 2016; Flam et al, 2022; Voros et al, 2018; Yurista et al, 2021). We found no DA ketone bodies (3-hydroxybutyrate and acetoacetic acid) in the myocardium of end-stage HF patients. Instead, we found protein downregulation of the intracellular monocarboxylate (ketone body) transporter 1 (SLC16A1), 3-hydroxybutyrate dehydrogenase (BDH1), and mitochondrial acetyl-CoA acetyltransferase (ACAT1) in ICM-DM, which are required for ketone metabolism to acetyl-CoA.

Proteins that catalyse BCAA metabolism to acetyl-CoA and succinyl-CoA, an alternative pathway to the citrate cycle, were also downregulated in HF or exclusively in ICM-DM. This suggests downregulation of BCAA metabolism in HF, particularly in ICM-DM, and may explain the upregulation of myocardial BCAAs in HF, as previously indicated in end-stage NICM (Flam et al, 2022) whereby increased tissue BCAA may impair insulin sensitivity (Uddin et al, 2019).

Myocardial Isoleucine leucine (all HF), leucine (ICM and NICM-DM), and valine (ICM) were upregulated in ICM-DM. Increased plasma BCAA concentrations have been found in obese and type 2 DM patients, and DM-induced rodents with cardiomyopathy, and have been suggested to impair insulin-stimulated glucose uptake via the mTORC1 pathway (Lee et al, 2016a; Lynch and Adams, 2014; Newgard et al, 2009; Tai et al, 2010; Tobias et al, 2018; Tremblay et al, 2005). Increased plasma and myocardial BCAAs have also been found in a diabetic cardiomyopathy mouse model with reduced expression of BCAT2, where pyridostigmine induced BCAA metabolism and clearance and attenuated the formation of myocardial scar tissue (Yang et al, 2021).

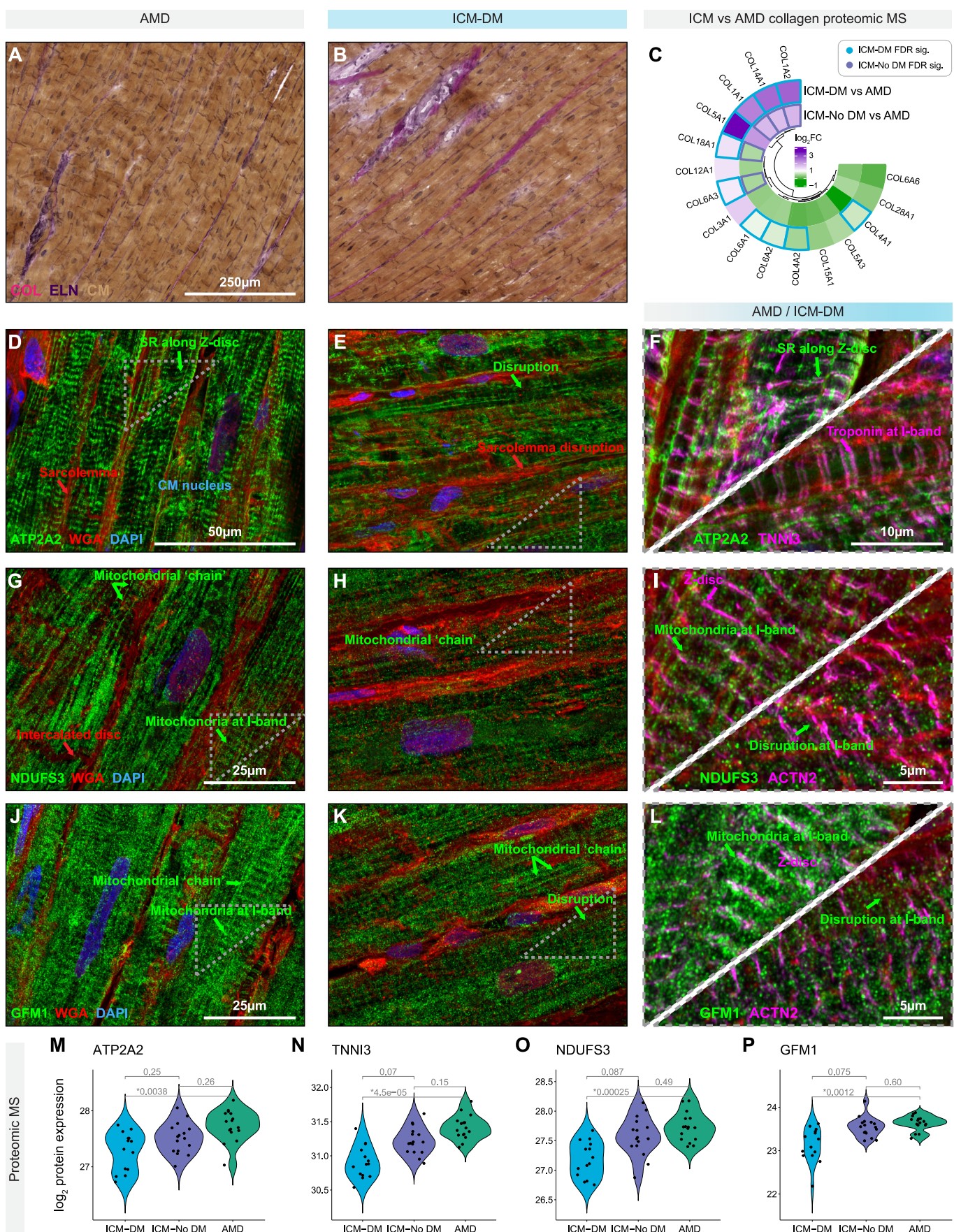

◄ Figure 6. **Histochemical visualisation of select key proteins in cryopreserved human left ventricular myocardium of ischaemic cardiomyopathy with diabetes (ICM-DM) and healthy age-matched donor (AMD) tissues.**

(A, B) Verhoeff-Van Gieson stain of an AMD and ICM-DM with collagens (COL, magenta), elastin (ELN, purple), cardiomyocytes (CM, brown), and nuclei (blue) (C), ICM-DM and ICM-No DM $\log_2$ fold change (FC) vs AMD, quantified collagen proteins from proteomic mass spectrometry (MS) where Benjamini–Hochberg false discovery rate (FDR) adjusted significantly differentially expressed proteins are indicated. (D–L) Immunofluorescent confocal microscopy images depicting qualitative differences between AMD and ICM-DM in the labelled protein of interest (green) with a labelled reference protein (magenta). Membranes were stained with fluorophore-conjugated wheat germ agglutinin (WGA, red), and nuclei were stained using DAPI (blue). All images are 4.5-µm-thick Z-stacks, deconvolved using Huygens Professional, and compressed into a two-dimensional image using Fiji/ImageJ Maximum Intensity Projections. (M–P) Proteomic MS $\log_2$ transformed quantification of immunohistological proteins of interest, ATP2A2, TNNI3, NDUFS3, and GFM1, in ICM-DM, ICM-No DM, and AMD groups. FDR-adjusted P values were (*FDR < 0.05) between groups. Differential analyses for (M–P) were performed using a moderated t test with the limma package (version 3.56.2) in R (version 4.3.1) following $\log_2$ transformation. ICM-DM $n = 14$, ICM-No DM $n = 16$, AMD $n = 20$.

## Mitochondria, the citrate cycle, and oxidative phosphorylation

Our finding of the downregulation of OXPHOS subunits, particularly CI, in HF and has been reported previously in rodent and human HF and diabetes suggesting a more common HF pathway (Chouchani et al, 2014; Forte et al, 2019; Karamanlidis et al, 2013; Liu et al, 2014; Montaigne et al, 2014; Sebastiao et al, 2022). We found subunits of CI, CIII, CIV, and CV (Fig. 4A,E) with substantially greater significance and FCs in ICM-DM vs AMD relative to the other HF conditions. However, our modelling simulation incorporating CI changes alone demonstrated that reduced FCs are not a rate-limiting factor for sufficient NADH to $NAD^+$ conversion/proton gradient, as suggested before (Lucas and Szweda, 1999; Valsecchi et al, 2012; Wu et al, 2007), but a limited capacity may be detrimental under stress (Karamanlidis et al, 2013; Stride et al, 2013). Furthermore, downregulated CI may result in other pathological consequences such as increased oxidative stress, hypertrophy, fibrosis, and glycolysis, and cell death via the mTOR signalling pathway (mTOR pathway proteins exclusively upregulated in ICM-DM) (Balsa et al, 2020; Chouchani et al, 2014; Karamanlidis et al, 2013; Perez-Gomez et al, 2020). Evidence of this effect in our observations was seen in significant upregulation of the $NAD^+$/NADH ratio in HF paired with the downregulation of $NAD^+$ consumer sirtuin 3 (SIRT3) (Xie et al, 2020). SIRT3 is a regulator of mitochondrial acetyl-CoA and OXPHOS via its deacetylation activity (Ahn et al, 2008; Cimen et al, 2010; Cui et al, 2017; Kane and Sinclair, 2018; Karamanlidis et al, 2013; Koentges et al, 2015; Murugasamy et al, 2022; Palomer et al, 2020; Sundaresan et al, 2016). Our findings are in apparent contrast to previous studies reporting reduced $NAD^+$/NADH in HF and in diabetes, mostly in hypertrophy HF rodent models, which may not translate to patients expressing end-stage HF prior to heart transplantation who do not have concentric cardiac hypertrophy but rather a dilated phenotype (Berthiaume et al, 2019; Chiao et al, 2021; Karamanlidis et al, 2013; Lee et al, 2016b). As high $NAD^+$/NADH ratios have been coupled with tissue diacylglycerol accumulation via the NADH-dependent actions of glycerol-3-phosphate dehydrogenase, both of which are downregulated in ICM-DM, our findings remain consistent (Fan et al, 2020; Mracek et al, 2013).

OXPHOS was further impaired in ICM-DM by the impairment of $FADH_2$ oxidation in three pathways: (1) Oxidation via CII and the citrate cycle; (2) FAO $FADH_2$ synthesis by the downregulation of acyl-CoA dehydrogenases; and (3) Oxidation by GPD2 via glycerol metabolism (Alcazar-Fabra et al, 2016; Mracek et al, 2013).

This was supported in immunofluorescence microscopy by the apparent reduction of unbound flavins (Chorvat et al, 2004; Chorvat et al, 2005; Kunz and Gellerich, 1993; Romashko et al, 1998; Wust et al, 2015). In addition, CoQ mediates the essential electron transfer from multiple sources (NADH, $FADH_2$) between the biochemical actions of CI–CIII and is also a source of reactive oxygen species, while cytochrome C facilitates electron transfer to CIV (Hernansanz-Agustin and Enriquez, 2021). We show CoQ subunits to be downregulated specifically in ICM-DM, while COQ9 was downregulated in DM.

Increased ROS formation from OXPHOS is principally due to pathological reverse electron transfer and it is associated with the metabolic dysfunction of DM (Hernansanz-Agustin and Enriquez, 2021; Kaludercic and Di Lisa, 2020). The gene sets of iron–sulfur clusters, which mediate electron transfer in CI–CIII, were downregulated specifically in DM (Read et al, 2021). Mitochondrial isocitrate dehydrogenase and malic enzyme 3 (ME3), which synthesise mitochondrial NADPH, were found to be downregulated in ICM-DM. Though G6PD was upregulated, it has been shown that there is no direct transport of NADPH from the cytosol to the mitochondria (Niu et al, 2023). In concordance with our results, it has been found that reduction of CI results in reduced mitochondrial NADPH production and increased oxidative stress (Balsa et al, 2020).

Regarding intracellular ATP transport, we show downregulation of the ATP/ADP antiporter, which transports ATP into the cytosol. We also found that kinases which increase ATP availability on demand, catalyse the formation of phosphocreatine, bind ATP, and act as an ATP energy buffer, were downregulated in HF, particularly in ICM-DM, which is consistent with previous studies which also associated downregulation of creatine kinase with a reduced OXPHOS capacity in HF, as well as overall reduced ATP availability (Stride et al, 2013; Weiss et al, 2005).

Mitochondrial dysfunction is not exclusive to diabetes and HF but also represented in other chronic diseases (Diaz-Vegas et al, 2020). Yet, common protein markers used to predict mitochondrial volume/activity have been shown to be inadequate across different organs and in disease (Kumar et al, 2019; Larsen et al, 2012; McLaughlin et al, 2020; Morgenstern et al, 2021). We propose SUCLG1, CKMT2, FH and ECHS1 as potential myocardial mitochondrial protein biomarkers, as indicated by a reduced volume in ICM-DM which was immunobiologically verified by qualitatively reduced mitochondria.

Mitochondria are normally arranged in a regular, repeating pattern between myofibrils and between Z-discs and the cardiomyocyte cytoskeleton (Adams et al, 2023; Kuznetsov et al, 2020;

Lyra-Leite et al, 2021; Vendelin et al, 2005). We also observed disorganisation of mitochondria along the I-band, like the studies showing more numerous yet smaller mitochondria with cristae loss in ICM (Chen et al, 2009; Liu et al, 2014) and diabetic (Dabkowski et al, 2009) rodent studies. However, we did not observe upregulation in fission-promoting proteins in HF in concordance with a human diabetes study which also reported no significant correlation between mitochondrial function and BMI (Li et al, 2020a; Montaigne et al, 2014). To our knowledge, there has been no descriptions differentiating ICM with and without DM.

## Compromised critical contractile complexes

Considering the downregulation of critical contractile complexes, we found that cardiac sarcoplasmic reticulum $Ca^{2+}$ handling proteins ATPase and RYR2 were downregulated in ICM-DM only. These have been identified to be reduced in HF previously, particularly the cardiac isoform of ATP2A2/SERCA2 wherein its affinity to $Ca^{2+}$ and its expression were reduced (Abe et al, 2002; Arai et al, 1993; Flesch et al, 1996; Lipskaia et al, 2014; Muslimova et al, 2020; Ohkusa et al, 1997; Studer et al, 1994; Vangheluwe et al, 2007; Vetter et al, 2002). In addition, $Na^+/Ca^{2+}$ exchanger SLC8A1 ($Ca^{2+}$ efflux) was downregulated specifically in ICM-DM, which contrasts with a study identifying increased left ventricular $Na^+/Ca^{2+}$ exchanger expression and activity in patients with end-stage HF (Reinecke et al, 1996). SLC8A1 inhibitors have been proposed as a therapeutic target for HF as SAR340835-treated HF dogs showed an improved stroke volume and sympathovagal balance (Pelat et al, 2021). However, the positive effects in acute HF induced in animal models may not translate to chronic human end-stage HF (Pott et al, 2011). We also identified the down-regulation of CASQ2 ($Ca^{2+}$ binding) in DM, and the upregulation of PLN, an inhibitory protein of ATP2A2. These changes of poor $Ca^{2+}$ handling impede left ventricular relaxation and influence the rate of contraction in all end-stage HF conditions, but mostly in ICM-DM (Abe et al, 2002; He et al, 1997; Kranias and Hajjar, 2012).

Confocal microscopy revealed disorganisation of ATP2A2 along the I-band of the sarcomere in ICM-DM, co-localising with TNNI3. TNNI3 was downregulated in ICM-DM and NICM, suggesting further contractile impairment which is consistent with its release and elevation in the circulation of patients with advanced HF (Bollen et al, 2017; James et al, 2000; Missov et al, 1997). Furthermore, it has recently been identified that nuclear transloca-tion TNNI3 positively regulates ATP2A2 expression and $Ca^{2+}$ uptake (Lu et al, 2022).

ALDOA has been identified as a critical enzyme promoting glycolysis and as an F-actin binding protein at the I-band and a positive regulator of $Ca^{2+}$ ryanodine receptors (Clarke and Morton, 1976; Kraft et al, 2000; Kramerova et al, 2008; Seo et al, 2006). ALDOA is in low concentrations in post-MI patients' plasma wherein an increased expression may have a protective role to alleviate oxidative stress and impair cardiomyocyte apoptosis in ischaemic conditions via the VEGF-NOTC1-JAG1 pathway (Luo et al, 2021). However, overexpression of ALDOA has also been attributed to cardiac hypertrophy development (Li et al, 2018b). Our study identifies that ALDOA co-localises at the I-band/Z-disc of human cardiomyocytes and is downregulated and dislocated in ICM-DM, suggesting DM as a specific driver.

## Regulators of the extracellular matrix

The expression of AEBP1, a protein known for its positive regulation of ECM organisation and fibrosis-related proteins, including COL1A1 and COL1A2, which exhibited RNA–protein correlation/co-regulation in ICM-DM (Blackburn et al, 2018), has been demonstrated to increase in response to hyperglycaemia, high fructose, and lipid concentrations in human hepatocytes (Gerhard et al, 2019). The expression of AEBP1 has also been linked to diabetes (Tao et al, 2021). A AEBP1 has also been shown to act as a transcription factor that negatively regulates the gene expression of fatty acid binding protein, FABP4 (Lyons et al, 2006). However, overexpression of AEBP1 in mice on a high-fat diet resulted in increased obesity (Ro et al, 2007; Zhang et al, 2005).

Myocardial upregulation of the ECM and COL deposition via myofibroblasts are common maladaptive feature in ICM and NICM (Fan et al, 2012; Frangogiannis, 2019; Li et al, 2018a; Liu et al, 2017; Travers et al, 2016). We identified ICM-DM upregulation of fibroblast activation protein alpha (FAP) transcript. Fibroblast integrin proteins, which promote ECM adhesion, can interact with POSTN, an ECM scaffolding protein, and stimulate angiogenesis, fibroblast proliferation and differentiation, and COL1 deposition by positively regulating TGFB1 and α-smooth muscle actin (Ashley et al, 2017; Deng et al, 2021; Nan et al, 2024; Qiao et al, 2024; Xu et al, 2021). In addition, dermatopontin (DPT) positively regulates ECM formation, including fibronectin and COL deposition and fibrillogenesis, which may promote maladaptive reverse remodelling in ICM and NICM (Lu et al, 2021; Okamoto and Fujiwara, 2006; Zhao et al, 2022). Our study found DPT was upregulated in all HF conditions with a greater than two-fold increase in ICM-DM relative to other HF conditions. This aligns with the previous finding of DPT and fibrosis upregulation in adipocytes of individuals with obesity and obesity-associated type 2 DM (Unamuno et al, 2020).

ECM secretory proteins fibulin 5 (FBLN5) and microfibril-associated protein 4 (MFAP4, RNA–protein-co-regulated) were upregulated in all HF conditions, most in ICM-DM, and were recently identified as HF-associated serum proteins independent of traditional risk factors suggesting plasma biomarkers of left ventricular origin (Dataset EV74) (Shah et al, 2024; Uhlén et al, 2019).

Taken together, we hypothesise a pathological feedback loop in the ICM-DM myocardium. ICM induced mitochondrial stress and an increase in ROS, promoting lipogenesis may inhibit carnitine transport into the mitochondria and FAO, while also promoting the breakdown of lipid droplets in response to energy deprivation and the inhibition of adaptive glucose influx because of diabetes. The resulting intracellular lipid overload may contribute to impaired cardiac contraction, induced fibrosis, and an efflux of fatty acids from the cardiomyocytes. Further energy deprivation may be possible due to the apparent downregulation of glycolysis, ketone and amino acid metabolism, and dysfunction of mitochondria.

## Limitations

This study is not without limitations. Efforts were made to control variables between conditions (age, macroscopic scar tissue, etc) to limit false results, including separate BMI and sex analyses where control was not possible. Acquisition and comparison of non-cardiomyopathy myocardium with diabetes was not possible. Not all proteins were quantified in this study, particularly

low-abundance proteins, which may also have resulted in a lower RNA–protein correlation.

## Conclusion

To our knowledge, this is the most comprehensive multi-omics resource on the molecular influence of DM in human end-stage ICM and NICM to date. Our results indicate that the cardiometabolic dysfunction of HF is further confounded and exacerbated in the presence of diabetes.

## Methods

### Reagents and tools table

| Reagent/resource | Reference or source | Identifier or catalogue number |
|---|---|---|
| **Experimental models** | | |
| Human left ventricular myocardium | This study | N/A |
| **Antibodies** | | |
| Alexa Flour 647-conjugated wheat germ agglutinin | Molecular Probes | W32466 |
| Alexa Fluor 488 donkey anti-rabbit IgG | Thermo Fisher Scientific | ab15065 |
| Anti-ACTN2 mouse IgG | Sigma-Aldrich | A7732 |
| Anti-ALDOA rabbit IgG | Sigma-Aldrich | HPA004177 |
| Anti-ATP2A2/SERCA2 mouse IgG | Thermo Fisher Scientific | MA3919 |
| Anti-GFM1 rabbit IgG | Sigma-Aldrich | HPA061405 |
| Anti-NDUFS3 rabbit IgG | Abcam | ab177471 |
| Anti-TNNI3 (cardiac troponin I) rabbit IgG | Abcam | ab47003 |
| DyLight 550 IgG donkey anti-mouse IgG | Abcam | ab9879 |
| **Chemicals, enzymes, and other reagents** | | |
| Acetonitrile (HPLC grade) | Thermo Fisher Scientific | A955-4 |
| Ammonium formate | Sigma-Aldrich | 70221 |
| Chloroacetamide | Sigma-Aldrich | C0267 |
| Chloroform (HPLC grade) | Thermo Fisher Scientific | C606SK-1 |
| DAPI | Thermo Fisher Scientific | 62248 |
| DPX | Sigma-Aldrich | 06522 |
| Formic acid (LC-MS grade) | Thermo Fisher Scientific | A117-50 |

| Reagent/resource | Reference or source | Identifier or catalogue number |
|---|---|---|
| Methanol (HPLC grade) | Thermo Fisher Scientific | A456-4 |
| Mito-ID | Enzo Life Sciences | ENZ-51022-0100 |
| Pierce™ BCA Protein Assay | Thermo Fisher Scientific | 23225 |
| ProLong Diamond antifade mountant | Thermo Fisher Scientific | P36970 |
| Qubit™ dsDNA High Sensitivity Quantification Assay Kit | Thermo Fisher Scientific | Q32851 |
| Sodium deoxycholate | Sigma-Aldrich | 30970 |
| Superfrost Plus slides | Menzel, Thermo Fisher Scientific | J1810AMNZ |
| TCEP | Thermo Fisher Scientific | 20490 |
| Tris-HCl pH 7.5 | Sigma-Aldrich | T5941 |
| Triton™ X-100 | Sigma-Aldrich | T8787 |
| TRIzol | Invitrogen | 15596026 |
| Trypsin (sequencing grade) | Promega | V5111 |
| Van Gieson's stain | This study | N/A |
| Verhoeff's Iron Haematoxylin solution | This study | N/A |
| **Software** | | |
| Adobe Illustrator | Adobe | Version 27.8 |
| BioRender | https://biorender.com | N/A |
| clusterProfiler | https://bioconductor.org/packages/clusterProfiler | Version 4.8.1 |
| DeepVenn | https://www.deepvenn.com/ | N/A |
| DIA-NN | https://github.com/vdemichev/DiaNN | Version 1.7 |
| Gene Ontology (GO) | https://geneontology.org/ | Version 3.17.0 |
| ggplot2 | https://github.com/tidyverse/ggplot2 | Version 3.4.2 |
| GraphPad Prism | GraphPad | Version 9.5.1 |
| Huygens Professional | Scientific Volume Imaging | Version 22.10.0p1 |
| ImageJ | ImageJ | Version 1.54f |
| KEGG | https://www.genome.jp/kegg/ | Version 102.0 |
| KEGGREST | https://github.com/Bioconductor/KEGGREST | Version 1.40.0 |
| limma | https://github.com/gangwug/limma | Version 3.56.2 |
| LipidSearch | Thermo Fisher Scientific | Version 4.2 |
| Microsoft Excel | Microsoft | Version 2309 |
| MitoCarta3.0 | Rath et al, (2021) | N/A |

| Reagent/resource | Reference or source | Identifier or catalogue number |
|---|---|---|
| NIS-elements | Nikon | Version 4.0 |
| PANTHER | http://www.pantherdb.org | Version 17.0 |
| R | R | Version 4.3.1 |
| Rsubread | https://bioconductor.org/packages/release/bioc/html/Rsubread.html | Version 2.2.1 |
| SCIEX OS | AB Sciex | Version 1.7.0.36606 |
| scMerge | https://github.com/SydneyBioX/scMerge | Version 1.16.0 |
| SRplot | https://www.bioinformatics.com.cn/en | N/A |
| STRING | https://string-db.org | Version 12.0 |
| ZEN Blue | Zen | Version 3.3 |
| **Other** | | |
| Agilent 1260 Infinity LC system | Agilent Technologies | N/A |
| Agilent High Sensitivity D1000 Tapestation system | Agilent | N/A |
| Atlantis® HILIC column | Waters | N/A |
| Axio Scan.Z1 | Zeiss | N/A |
| Bioanalyzer | Agilent | N/A |
| C18 analytical column (1.9 μm) | Dr. Maisch GmbH | N/A |
| C18 HPLC column | Thermo Fisher Scientific | N/A |
| Cryotome FSE | Thermo Fisher Scientific | N/A |
| LabChip GX Touch | Revvity | N/A |
| NanoDrop | Thermo Fisher Scientific | N/A |
| Nikon C2plus confocal microscope | Nikon | N/A |
| NovaSeq 6000 | Illumina | N/A |
| Q-Exactive Fusion Lumos MS | Thermo Fisher Scientific | N/A |
| Q-Exactive HF-X MS | Thermo Fisher Scientific | N/A |
| QTRAP5500 Mass Spectrometer | AB Sciex | N/A |
| Savant SC210 SpeedVac | Thermo Fisher Scientific | N/A |
| Shiny App (Human-Heart-ALL) | The University of Sydney | http://shiny.maths.usyd.edu.au/Human-Heart-ALL/ |
| SpeedVac SPD120 | Thermo Fisher Scientific | N/A |
| ThermoMixerC | Eppendorf | N/A |
| TissueLyser LT | Qiagen | N/A |
| XBridge™ Amide column | Waters | N/A |

## Methods and protocols

### Acquisition of human myocardium

Pre-mortem human myocardium was acquired from the outer anterior and lateral walls of the left ventricle from explanted hearts of end-stage HF (ICM-DM, ICM-No DM, NICM-DM, and NICM-No DM) patients undergoing heart transplantation surgery, and from healthy/non-pathological donors whose hearts could not be viably used for heart transplantation due to logistical and compatibility limitations at the time. Procurement was performed at St Vincent's Hospital Sydney as previously described (Cao et al, 2019; Crossman et al, 2017; Lal et al, 2015; Lal et al, 2016; Lange et al, 2016; Li et al, 2020b; Mamidi et al, 2017; Mollova et al, 2013; Polizzotti et al, 2015; Sequeira et al, 2015; van Heesch et al, 2019). Hearts were perfused in cardioplegia, placed on wet ice, and myocardial samples were immediately snap-frozen on-site in liquid nitrogen (−196 °C) within 40 min after aortic cross-clamp such that the tissue was not post-mortem and that high-quality RNA was preserved (Lal et al, 2015; Lal et al, 2016; van Heesch et al, 2019). Samples were then stored long-term at −192 °C at the Sydney Heart Bank. Visible myofibrotic scar tissue, such as infarct-affected regions, were avoided. Pathology (HF) of myocardial tissue was histologically determined by hospital anatomical pathology. Diabetic patients were using Metformin and insulin and there were no significant differences in their diabetic care. Informed consent was obtained before the collection of all tissue. All human myocardial donors were Caucasian. This was not by selection, but rather due to tissue availability. The methods of procurement, storage, and use of donated human myocardium were approved by the Human Research Ethics Committee at The University of Sydney (USYD 2021/122). Experiments conformed to the principles set out in the WMA Declaration of Helsinki and the Department of Health and Human Services Belmont Report.

Dissemination of individual-level patient data is not permissible beyond condition/phenotype, age, sex, diabetes/no diabetes, and BMI. However, clinical and demographic data for each quantitative method has been provided in aggregate form (Datasets EV1, EV43, and EV55). Clinical information was not available for every individual. Patient de-identified study IDs and the experiments each individual contributed to is available in Dataset EV59.

### Proteomics

Cryopreserved human myocardium fragments, lacking macroscopic myofibrotic scar tissue, were crushed and powderised with a mortar and pestle at −196 °C, and weighed to ~10 mg. The powder was suspended in 4% sodium deoxycholate and 100 mM Tris-HCl pH 7.5 and lysed in the ThermoMixerC (Eppendorf) at 95 °C for 10 min. The homogenate was spun at 14,000 × g for 10 min and the supernatant collected. Stock protein concentration was quantified by performing a bicinchoninic acid (BCA, Pierce) assay and concentrations were corrected to 0.4 ng/μL. Protein was reduced in 10 nM TCEP and 40 nM chloroacetamide and denatured at 95 °C for 10 min. A trypsin digest was then performed overnight at 37 °C. Peptides were prepared for mass spectrometry as described previously (Harney et al, 2019). A pooled sample was formed and was separated offline with high-pH RP fractionation. An acquity UPLC M-Class CSH C18 130 Å pore size, 300 μm × 150 mm

column, and 1.7 μm internal diameter (Waters) was used to fractionate peptides. Separation was performed over 30 min, concatenating 96 fractions into 16 wells using buffer A; 2% acetonitrile and 10 mM ammonium formate (pH 9), and buffer B; 80% acetonitrile and 10 mM ammonium formate (pH 9). Fractionates were dried and resuspended in 5% formic acid. Peptides were directly injected into a C18 (Dr. Maisch, Ammerbuch, Germany, 1.9 μm) 30 cm × 70 cm column with a 10 μm pulled tip integrated online to a nanospray ESI source. Separation involved buffer A; 0.1% formic acid in LC-MS grade water, and buffer B; 80% acetonitrile and 0.1% formic acid in MS grade water. Peptides were differentiated from a gradient of 5–40% of buffer B over 2 h with a flow rate of 300 nL per minute and were ionised by electrospray ionisation at 2.3 kV. A Q-Exactive Fusion Lumos mass spectrometer (Thermo Fisher Scientific) was used to conduct MS/MS analysis with 27% normalised HCD collision energy for fragmentation at the Sydney Mass Spectrometry facility, the University of Sydney. Spectra were procured in a data-independent acquisition using 20 variable isolation windows. RAW data files including the high-pH fractions were analysed using the integrated quantitative proteomics software DIA-NN (Demichev et al, 2020) (version 1.7). The database provided to the search engine for identification contained the Uniprot human database downloaded on the 5th of May, 2020. FDR was set to 1% of precursor ions. Both remove likely interferences and match between runs were enabled. Trypsin was set as the digestion enzyme with a maximum of 2 missed cleavages. Carbamidomethylation of Cys was set as a fixed modification and oxidation of Met was set as variable modifications. Retention time-dependent profiling was used, and the quantification setting was set to any LC (high accuracy). Protein inference was based on genes. The neutral network classifier was set to double-pass mode. The MaxLFQ algorithm was used for label-free quantitation, integrated into the DIA-NN environment (Cox and Mann, 2008; Cox et al, 2011).

Normalised and log$_2$ transformed proteomic data per individual is available in Dataset EV60.

### Metabolomics

Cryopreserved human myocardium was powdered as described in the proteomic MS methods and weighed to 50 mg. Cells were lysed and homogenised with steel balls in methanol/chloroform (2:1; v/v, HPLC grade) using the TissueLyser LT (Qiagen) and kept on dry ice between cycles (3–5 ×1 min). Equal volumes of chloroform then water (HPLC grade) were added to promote protein precipitation. Protein and debris were pelleted at 14,000 rpm at 4 °C for 20 min and the metabolite-containing supernatant was collected. The SpeedVac SPD120 (Thermo Fisher Scientific) was used to dry the solution under nitrogen steam. Compounds were resuspended in the acetonitrile/methanol/formic acid (75:25:0.2; v/v/v, HPLC grade; Thermo Fisher Scientific) for the HILIC analysis, and acetonitrile/methanol (25:25; v/v/v, HPLC grade) for the AMIDE analysis. Targeted metabolite profiling was performed using deuterated internal standards to determine mass spectrometry (MS) multiple reaction-monitoring transitions, declustering potentials, collision energies and chromatographic retention time, as described previously (Koay et al, 2021a; Koay et al, 2021b). HILIC and AMIDE mass spectrometry analyses were performed via liquid chromatography-tandem mass spectrometry (LC-MS/MS) using an

Agilent 1260 Infinity liquid chromatography (Santa Clara, CA, USA) system coupled to a QTRAP5500 mass spectrometer (AB Sciex, Foster City, CA, USA) at the Sydney Mass Spectrometry facility, the University of Sydney. Positive and negative ion modes were used to separate polar compounds in hydrophilic interaction liquid chromatography (HILIC) mode using an Atlantis® HILIC column (Waters), and an XBridgeTM Amide column (Waters), respectively, which allow the separation of metabolites of different properties, as previously described (Koay et al, 2021a; Koay et al, 2021b). Samples were randomised across three sequential batches. Pooled metabolite extracts formed from each sample were included after every 10 study samples in the sample queue to detect temporal dips in instrument performance following the analysis. SCIEX OS (AB SCIEX, version 1.7.0.36606) was later used for multiple reaction-monitoring Q1/Q3 peak integration of the raw data files whereby the abundance was quantified as the peak area of a metabolite. QC was performed removing metabolites with poor quantification or to select between metabolites measured in both negative and positive modes. Metabolite abundances were normalised in R (version 4.3.1.) using scMerge (version 1.16.0) (Lin et al, 2019).

Normalised and log$_2$ transformed metabolomic data per individual is available in Dataset EV61.

### Lipidomics

Cryopreserved human myocardium was powdered as described in the proteomic MS methods and weighed to 20 mg. Internal standards were added to each sample in 250 μL methanol (MS grade) containing 0.01% (w/v) butylhydroxytoluene (BHT); 2 nmoles PC(19:0/19:0); 1 nmole each of SM(d18:1/12:0), Glu-Cer(d18:1/12:0), Cer(d18:1/17:0), PS(17:0/17:0), PE(17:0/17:0), PA(17:0/17:0), PI(d7-18:1/15:0), PG(17:0/17:0), CL(14:0/14:0/14:0/14:0), and TG(17:0/17:0/17:0); 0.5 nmoles each of DG(d7-18:1/15:0), CholE(17:0), LPC (17:0), LPE(17:1), and AcCa(d3-16:0); and 0.2 nmole each of Sph(d17:1), S1P(d17:1), LacCer(d18:1/12:0), and MG(d7-18:1). Samples were then homogenised using steel balls in the TissueLyser LT (Qiagen) and kept on wet ice between cycles (3–5 ×1 min). Methyl-tert-butyl-ether (MTBE, 1 mL) was added to the homogenate and samples sonicated at 4 °C for 30 min. Phase separation was induced by adding 250 μL of water (MS grade), vortexing and spinning at 2000 × g for 5 min. The upper organic phase was extracted in 5-mL glass tubes. Extraction was performed twice more; sonicated at 4 °C for 30 min in 500 μL MTBE and 150 μL methanol, then vortexed and spun at 2000 × g for 5 min after adding 150 μL water. Lipids were dried under vacuum in a Savant SC210 SpeedVac (Thermo Fisher Scientific) and reconstituted in 400 μL 80% methanol/20% water/0.1% formic acid containing 0.01% (w/v) BHT. Lipids were then analysed via liquid chromatography-tandem mass spectrometry (LC-MS/MS) at the Sydney Mass Spectrometry facility, the University of Sydney. LC-MS/MS was performed using a Q-Exactive HF-X mass spectrometer (Thermo Fisher Scientific) coupled to a Vanquish HPLC with a 2.1 × 100 mm C18 HPLC column (Waters, 1.7-μm pore size) as previously described (Couttas et al, 2020) with amendments in the following. HPLC solvent A was 10 mM ammonium formate, 0.1% formic acid in acetonitrile:water (60:40), and solvent B was 10 mM ammonium formate, 0.1% formic acid in isopropanol:acetonitrile (90:10). A 27 min binary gradient at 0.28 mL/min was used: 0 min, 80:20 A/B; 3 min, 80:20 A/B; 5.5 min, 55:45 A/B; 8 min, 36:65 A/B;

13 min, 15:85 A/B; 14 min, 0:100 A/B; 20 min, 0:100 A/B; 20.2 min, 70:30 A/B; 27 min, 70:30 A/B. Data were acquired in full scan/data-dependent MS2 mode (full scan resolution 60,000 FWHM, scan range 220–1600 $m/z$). The sample order was randomised, and data was collected in both positive and negative mode for each sample. The ten most abundant ions in each cycle were subjected to MS2, with an isolation window of 1.4 $m/z$, collision energy 30 eV, resolution 17,500 FWHM, maximum integration time 110 ms and dynamic exclusion window 10 s. A solvent blank was used to created an exclusion list of background ions while an inclusion list was used for all internal standards. LipidSearch software (version 4.2, Thermo Fisher Scientific) was used for lipid annotation, chromatogram alignment, and peak integration. Lipid annotation required both accurate precursor ion mass (tolerance 5 ppm) and diagnostic product ions (tolerance 8 ppm). Individual lipids were expressed as ratios to the class-specific internal standard, then multiplied by the amount of internal standard to calculate molar amounts for each lipid. Lipid levels were expressed as nmoles/mg tissue.

Normalised and $\log_2$ transformed metabolomic data per individual is available in Dataset EV62. Lipid class names are available in Dataset EV63.

### RNA sequencing

Cryopreserved human myocardium was powdered as described in the proteomic MS methods and weighed to ~30 mg. Powderised myocardium was lysed and homogenised in 500 μL TRIzol (Invitrogen) using steel balls in the TissueLyser LT (Qiagen) and kept on dry ice between cycles (3–5 ×1 min). In all, 1-bromo-3-chloropopane (50 μL) was added to each sample and left to incubate for 5 min at RT, then spun at 14,000 × g at 4 °C for 15 min to induce phase separation. RNA-containing aqueous phase was transferred to a sterile tube and an equal volume of isopropanol was added, mixed by inverting, and left for 1 h at RT. RNA was pelleted at 14,000 × g at 18 °C for 15 min, supernatant discarded. RNA was then washed in 4 °C 70% ethanol twice discarding supernatant after spinning at 14,000 × g at 4 °C for 10 min and left to dry at RT. RNA was DNase-treated for 30 min at 37 °C and washed again. Pelleted RNA was reconstituted in 15 μL 4 °C nuclease-free diethyl pyrocarbonate (DEPC). Concentration and quality were tested on NanoDrop (Thermo Fisher Scientific) to a standard of 260 nm/280 nm = 1.8–2.0 and 260 nm/230 nm 1.7–2.2. RNA integrity was assessed using an RNA Nano Chip on an Bioanalyzer (Agilent). RNA-seq libraries were prepared with Illumina Stranded Total RNA prep Ligation with Ribo Zero Plus (100 ng input and 11 PCR cycles) according to the manufacturer's instructions. Quality checks were performed using the Qubit dsDNA High Sensitivity Assay Kit (Thermo Fisher Scientific) and the LabChip GX Touch (Revvity). The final sequencing pool was quantified using the Qubit dsDNA High Sensitivity Assay Kit after pooling all libraries equimolar into a single library pool. Sizing was checked using the Agilent High Sensitivity D1000 Tapestation system. The RNA-seq libraries were sequenced using a paired-end 250 bp kit on a S4 flow cell of the NovaSeq 6000 (Illumina) with a final run concentration of 58 pM and 1% PhiX. The raw data was demultiplexed using bcl2fastq. Library preparation and sequencing were performed by the Ramaciotti Centre for Genomics, at the University of New South Wales, Australia. The quality of each RNA sequence (reversely stranded) was assessed using ShortRead, and the alignment was performed on paired-end sequences without trimming (Liao et al, 2019; Morgan et al, 2009). The alignment of raw sequences was performed on hg38 genome assembly (UCSC) with Rsubread (version 2.2.1) (Liao et al, 2019; Pan et al, 2019). Rsubread was also used for generating the gene count matrix. The gene counts were converted in Count Per Million (CPM), $\log_2$ transformed, and then normalised using voom (Law et al, 2014).

Normalised and $\log_2$ transformed RNA sequence data per individual is available in Dataset EV64.

### Statistics

Statistical analyses were performed on anonymised datasets. Samples were randomised during mass spectrometry acquisition and RNA sequencing library preparation to minimise batch effects prior to statistical analysis.

Differential abundance (DA) or differential expression analyses were performed on proteomic, metabolomic, lipidomic, and RNA-seq datasets using a moderated $t$ test with the limma package (version 3.56.2) in R (version 4.3.1) following $\log_2$ transformation. HF conditions (ICM and NICM) were compared with AMDs; Donors aged 47–65 years for ICM, and Donors aged 21–65 years for NICM. DA analyses were also performed for ICM-DM was also compared against ICM-No DM, and males against females. Molecules with missing values in more than 25% of samples were excluded. No imputation method was applied. Significant molecules were determined by controlling for False discovery rate at the 5% level (FDR < 0.05) using Benjamini–Hochberg FDR adjustment (FDR < 0.05) and ranked using the topTable function in limma. Differential analyses are located in Datasets EV2–EV20, EV44–EV46, EV52, EV56–EV58.

Correlation between RNA and protein (RNA–protein), BMI and molecule, and individual mitochondrial protein expression and the average MitoCarta3.0 protein expression were calculated using Pearson's correlation coefficient, whereas the correlation between male and AEBP1 fold change (FC, HF compared to AMD) was calculated using Spearman's correlation coefficient. Correlation between individual mitochondrial protein expression and the total average expression of the mitochondrial proteome (MitoCarta3.0) were also calculated via Pearson's correlation coefficient where the mitochondrial proteome was limited to proteins quantified and proteins expressed by all samples. Correlation was determined as ρ > 0.8 (Pearson, exceeded FDR), or as ρ > 0.8 and $P < 0.05$ (Spearman). Correlation analyses are located in Datasets EV37–EV42, EV48–EV51.

Gene Set Enrichment Analysis (GSEA) analyses were performed on normalised and transformed proteomic and metabolomic datasets using clusterProfiler (version 4.8.1) (Wu et al, 2021) in R where significance was determined as $P < 0.05$ (unadjusted). Protein enrichment was performed referencing the databases; Kyoto Encyclopedia of Genes and Genomes (KEGG, accessed 21 June 2022) (Kanehisa, 2019; Kanehisa et al, 2023; Kanehisa and Goto, 2000) via KEGGREST (version 1.40.0), MitoCarta3.0 (Rath et al, 2021), and Gene Ontology (GO, version 3.17.0) Biological process (BP) and Cellular Component (CC) (Aleksander et al, 2023; Ashburner et al, 2000). Selected mitochondrial and energetics-associated KEGG pathways and MitoCarta3.0 gene sets were manually annotated. The Kolmogorov-Smirnov statistic was used to compare the ranks of $P$ values of genes in the pathways/gene sets vs the uniform distribution using the "GSEA" function. GSEA GO

enrichment results were the child nodes of manually selected cytoskeletal, sarcomeric, and extracellular matrix-associated parent nodes; extracellular matrix organisation (GO:0030198), cardiac muscle cell differentiation (GO:0055007), myofibril assembly (GO:0030239), heart contraction (GO:0060047), cytoskeleton organisation (GO:0007010), extracellular matrix (GO:0031012), contractile fibre (GO:0043292), and cytoskeleton (GO:0005856). Significantly enriched protein pathways/gene sets/nodes were fully represented (NICM) or manually selected (ICM) in bubble plot figures. Metabolite enrichment was performed referencing on the KEGG database. clusterProfiler enrichment analyses are located in Datasets EV21–EV32.

Enrichment analyses of targeted genes were performed following and based on the FDR accepted proteomic and RNA-seq differential abundance analyses results using GO via PANTHER (version 17.0) (Thomas et al, 2022) and STRING (version 12.0) (Szklarczyk et al, 2019; Szklarczyk et al, 2023) where significance was determined as FDR < 0.05. STRING analysis (version 11.5) was also performed on ICM-DM RNA–protein correlated genes. GO via PANTHER analyses referenced BP, CC, and Molecular Function (MF) libraries where inputted data was direction specific (down or upregulated) for each analysis before combining the results. STRING analyses referenced GO (BP, CC, and MF) and KEGG databases. Significantly enriched nodes/pathways were manually selected for the figures. Targeted enrichment analyses are in Datasets EV33-EV36, EV47, EV53, and EV54.

### Graphics

Figure elements were generated in R using ggplot2 (version 3.4.2), clusterProfiler, GraphPad Prism (version 9.5.1), DeepVenn (Hulsen, 2022), Microsoft Excel (version 2309), Adobe Illustrator (version 27.8), at Biorender (https://app.biorender.com), and SRplot (https://www.bioinformatics.com.cn/en) (Tang et al, 2023). Figure 1A and the Synopsis graphics were created with BioRender.com. Molecules of the circular heat maps were clustered via complete-linkage hierarchical clustering, and $Log_2$ FCs were separated via Euclidean distance. RNA–protein scatter plots only plotted genes with a relative RNA expression ≥5. CC nodes of the GO chord plot were manually selected based on enrichment. UpSet plots were produced using FDR significant proteins/metabolites in HF vs AMD from the differential analyses. Elements were modified and formatted using Adobe Illustrator.

### Verhoeff-Van Gieson histochemistry and bright-field microscopy

Donor and ICM-DM cryopreserved pre-mortem human left ventricular myocardium was sectioned on the Cryotome FSE (Thermo Fisher Scientific) at −16 °C while in tissue freezing medium to 16 μm thickness and collected onto Superfrost Plus slides (Menzel, Thermo Fisher Scientific). Sections were fixed in methanol for 3 min at −20 °C, immersed in tap water. Elastin fibres and nuclei were stained in Verhoeff's Iron Haematoxylin solution; 2.8% filtered ethanol haematoxylin (w/v), 22% ferric chloride (v/v), 22% strong iodine, for 25 min then differentiated in 2% aqueous ferric chloride for 10 dips/seconds. Iodine was washed out in 95% ethanol for 30 s and collagen was then counterstained in Van Gieson's stain; 0.1% aqueous acid fuchsin, in 90% saturated picric acid (v/v), for 3 min. Sections were then dehydrated in ascending changes of ethanol, cleared in xylene, and mounted in DPX (Sigma-Aldrich).

Bright-field images (24 bid depth) were captured on ZEN blue (version 3.3) using the Zeiss Axio Scan.Z1 and a ×40 plan-apochromat objective with a numerical aperture (NA) of 0.95 on the Hitachi HV-F202SCL camera with an exposure time of 200 μs at the Sydney Microscopy and Microanalysis facility, the University of Sydney. Images were adjusted (brightness and contrast) to identical ranges and exported in Tag Image File Format (TIFF) using Fiji/ImageJ (Schindelin et al, 2012).

### Immunohistochemistry and confocal microscopy

Donor and ICM-DM sections were sectioned as described for Verhoeff-Van Gieson staining. Sections were encircled with a PAP pen and fixed in 10% neutral buffered formalin for 15 min then washed from free aldehyde groups in 0.2% glycine (w/v) in phosphate buffered saline (PBS). Slides were washed for 5 min three times in PBS between steps at 40 RPM. Tissue was permeabilised in 0.5% (v/v) Triton X-100 (Sigma-Aldrich) in PBS before non-specific binding sites were blocked in 5% (v/v) normal donkey serum and 5% acetylated bovine serum in PBS for 45 min at room temperature (RT) at 40RPM. Monoclonal primary antibodies (IgG) produced in rabbit labelling TNNI3 (ab47003, ~5 μg/mL, Abcam), NDUFS3 (ab177471, ~20 μg/mL, Abcam), GFM1 (HPA061405, ~16 μg/mL, Sigma-Aldrich), or ALDOA (HPA004177, ~10 μg/mL, Sigma-Aldrich) were applied to the sections in 10% blocking solution overnight at 4 °C at 40 RPM. Rabbit IgG were in-turn labelled with secondary fluorophore-conjugated Alexa Flour 488 IgG (ab15065, 5 μg/mL, abcam) overnight at 4 °C at 40 RPM. Mitochondria were also alternatively stained with Mito-ID (ENZ-51022-0100, 2 μL/mL, Enzo Life Sciences) in PBS for 15 min at RT at 40RPM. Co-labelling was similarly performed in sequence with mouse-produced monoclonal IgG labelling ATP2A2 (MA3919, ~10 μg/mL, Invitrogen) or ACTN2 (A7732, ~20 μg/mL, Sigma-Aldrich) and secondary IgG DyLight 550 (ab9879, 5 μg/mL, Abcam). Plasma/intramembrane glycoproteins were bound with Alexa Flour 647-conjugated wheat germ agglutinin (W32466, 5 μg/mL, Molecular Probes) for 1 h at RT at 40 RPM. Nuclei were stained with DAPI (62248, 1 μg/mL, Thermo Fisher Scientific) for 10 min at RT at 40 RPM. Sections were then mounted in ProLong Diamond antifade mountant (Thermo Fisher Scientific).

Z-stack images (12 bit depth) with a thickness of 4.5 μm and step size of 0.3 μm (16 images) were captured on NIS-elements (version 4.0) using a Nikon C2plus confocal microscope, a ×100 oil-immersed plan-apochromat objective with a NA of 1.45 and image parameters; 1.0 airy unit (AU), 1.3× digital zoom, 1/16 frames/sec, 2× frame-average, and a resolution of 2048 ×2048. Channels were imaged in sequence from long to short excitatory laser wavelengths: 640 nm (emission filter 650–700 nm), 561 nm (emission filter 570–620 nm), 488 nm (emission filter 500–550 nm), and 405 nm (emission filter 417–477 nm). Acquisition parameters, including laser power, gain, and offset were identical between donor and ICM-DM and no-primary IgG controls (ICM-DM). Images were acquired at the Sydney Microscopy and Microanalysis facility, the University of Sydney. All channels were deconvolved with Huygens Professional version 22.10.0p1 64b (Scientific Volume Imaging, The Netherlands, http://svi.nl), using the CMLE algorithm. Stain-specific templates were used between conditions to ensure identical deconvolution parameters (acuity, signal-to-noise ratio, iterations, and background value). Z-stacks were compressed into a two-dimensional image using Maximum Intensity

Projections, adjusted (brightness and contrast) to identical ranges and exported in OME- TIFF using Fiji/ImageJ (version 1.54 f) (Schindelin et al, 2012).

### In-silico modelling

To study the physiological consequences of reduced complex I (CI) activity, we employed a biophysical model of oxidative phosphorylation validated on rat cardiac myocytes (Beard, 2005; Vendelin et al, 2000; Vinnakota and Bassingthwaighte, 2004). We reduced the activity parameter corresponding complex I ($\mu_{C1}^a$) according to the equation

$$\mu_{C1}^a = \mu_{C1}^{a,0} + RT \ln \text{FC},$$

where $\mu_{C1}^{a,0} = -21.01$ kJ/mol is the baseline value of the activity parameter, $R = 8.314$ J/K/mol is the ideal gas constant, $T = 297.5$ K is the absolute temperature, and FC is the fold change in CI abundance relative to the baseline abundance. The parameters corresponding to the groups were:

- AMD: Baseline value (FC = 1.0).
- ICM-DM: -29.4% from baseline (FC = 0.706), consistent with fold changes in the proteomic data. The fold change was calculated from the mean of the fold changes (ICM-DM relative to AMD) in the CI subunits present in the proteomic data. CI subunits were defined using the HUGO Gene Nomenclature Committee (HGNC) groups "NADH:ubiquinone oxidoreductase core subunits" (https://www.genenames.org/data/genegroup/#!/group/1149) and "NADH:ubiquinone oxidoreductase supernumerary subunits" (https://www.genenames.org/data/genegroup/#!/group/1150).
- Low C1: -40% from baseline (FC = 0.6). This group was included to study the effects of reducing complex I beyond the fold changes seen in the proteomic data.

The model was simulated for each group by iteratively varying the ATP consumption parameter $\mu_{ATPase}^a$, leading to changes in the oxygen consumption rate $VO_2 = 0.5J_{C4}$. For each value of $\mu_{ATPase}^a$, the model was simulated for 1000 s to reach steady state.

## Data availability

Proteomic mass spectrometry raw data is available at ProteomeXchange via PRIDE (Perez-Riverol et al, 2022) with the project accession PXD052878 at http://proteomecentral.proteomexchange.org. Metabolomic mass spectrometry and lipidomic mass spectrometry raw data is available at Metabolomics Workbench (Sud et al, 2016) with the Study IDs ST003274 (metabolomics) and ST003275 (lipidomics) under Project PR002031 at https://doi.org/10.21228/M83529. https://doi.org/10.21228/M83529. Lipidomic mass spectrometry raw data is temporarily available at Metabolomics Workbench with the Study ID ST003275 and Project DOI PR002031 at https://doi.org/10.21228/M83529. RNA sequencing raw data is available at NCBI Gene Expression Omnibus (GEO) (Edgar, 2002) at https://www.ncbi.nlm.nih.gov/geo/query/acc.cgi?acc=GSE263297. Raw, processed, and adjusted confocal and bright-field microscopy images are available at https://zenodo.org/records/15788161. https://doi.org/10.5281/zenodo.15788161 (Hunter, 2025). Code for quantitative analyses is available at https://zenodo.org/records/14048213. https://doi.org/10.5281/

zenodo.14048213 (Zhang, 2025). Code for in-silico oxidative phosphorylation modelling is available at https://url.au.m.mimecastprotect.com/s/tHvhCL7EwMfQ2gxYOUBfviyxx2V?domain=github.com.

The source data of this paper are collected in the following database record: biostudies:S-SCDT-10_1038-S44321-025-00281-9.

## Peer review information

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

## Acknowledgements

We thank the patients and staff of St Vincent's Hospital, Sydney, and the Australian Red Cross Blood Service. We thank Emeritus Prof. Cris dos Remedios and the late Dr. Victor Chang AC, who established the Sydney Heart Bank. We thank Ben Crossett and Stuart Cordwell from Sydney Mass Spectrometry. This work was funded by the RT. Hall Trust (RQ112) and philanthropic donations to the University of Sydney. This work was also supported by the National Health and Medical Research Council (NHMRC) of Australia and the National Heart Foundation (NHF) of Australia. The contents of the published material are solely the responsibility of the individual authors and do not reflect the view of NHMRC or the NHF. JOS is supported by an NSW OHMR EMCR grant (G212785), NHF Vanguard Grant (105595), NHF Future Leader Fellowship (104853), and an NHMRC/MRFF CVD award (2024161). MP was supported by a Postdoctoral Research Fellowship from the School of Mathematics and Statistics, University of Melbourne. Metabolomics Workbench raw data availability is supported by NIH grant U2C-DK119886 and OT2-OD030544.

## Author contributions

**Benjamin Hunter**: Conceptualisation; Data curation; Software; Formal analysis; Investigation; Visualisation; Methodology; Writing—original draft; Project administration; Writing—review and editing. **Yunwei Zhang**: Data curation; Software; Formal analysis; Investigation; Visualisation; Methodology. **Dylan Harney**: Data curation; Software; Methodology. **Holly McEwen**: Data curation; Software; Methodology. **Yen Chin Koay**: Supervision; Methodology. **Michael Pan**: Software; Formal analysis; Investigation; Methodology. **Cassandra Malecki**: Data curation; Methodology. **Jasmine Khor**: Data curation. **Robert D Hume**: Supervision. **Giovanni Guglielmi**: Data curation. **Alicia Walker**: Visualisation. **Shashwati Dutta**: Visualisation. **Vijay Rajagopal**: Resources; Supervision. **Anthony Don**: Resources; Supervision; Methodology. **Mark Larance**: Resources; Supervision; Methodology. **John F O'Sullivan**: Resources; Supervision; Funding acquisition; Writing—review and editing. **Jean Y H Yang**: Resources; Supervision; Methodology. **Sean Lal**: Conceptualisation; Resources; Supervision; Funding acquisition; Investigation; Writing—original draft; Project administration; Writing—review and editing.

Source data underlying figure panels in this paper may have individual authorship assigned. Where available, figure panel/source data authorship is listed in the following database record: biostudies:S-SCDT-10_1038-S44321-025-00281-9.

## Disclosure and competing interests statement

The authors declare no competing interests.

# Expanded View Figures

**Figure EV1.  Sample visualisation of normalised and log$_2$ transformed proteomic, metabolomic and lipidomic mass spectrometry data from human myocardium in all conditions.** ▶

(**A–L**) Multidimensional scaling (MDS) plots, a tool to visualise high dimensional data in two dimensions, were used to iteratively separate samples (each point) in distance based on their dissimilarity of paired data (proteins, metabolites, and lipids; dimensions) calculated as the leading log$_2$ fold change (FC, average root-mean-square of the largest log$_2$ FCs). All protein/metabolite/lipid MDS plots are the same but vary in samples highlighted. (**A–C**) All conditions highlighted. (**D–F**) Heart failure (HF) conditions and healthy donors highlighted. (**G–I**) Ischaemic cardiomyopathy with diabetes (ICM-DM) and without diabetes (ICM-No DM) highlighted. (**J–L**) Non-ischaemic cardiomyopathy with diabetes (NICM-DM) and without diabetes (NICM-No DM) highlighted.

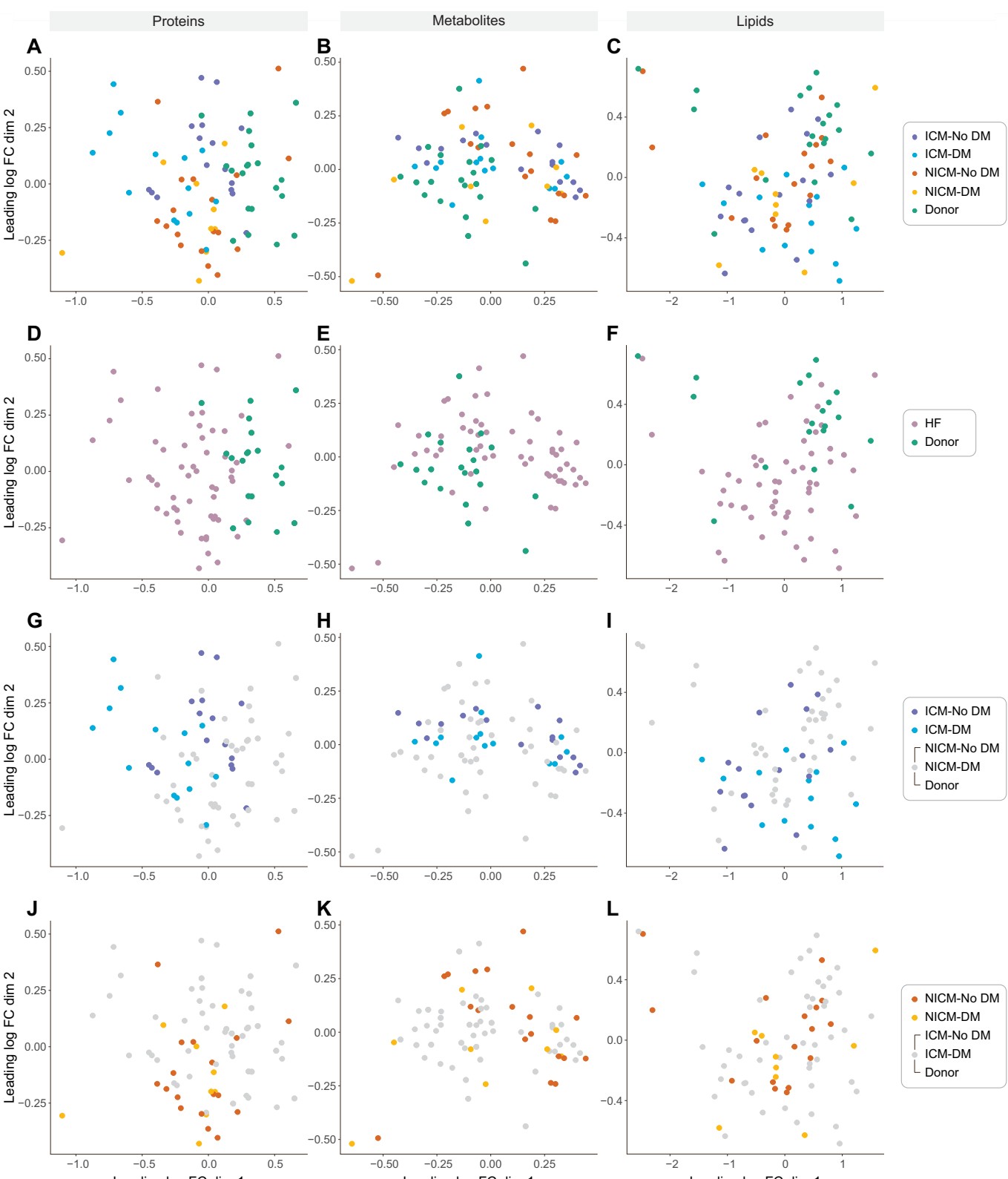

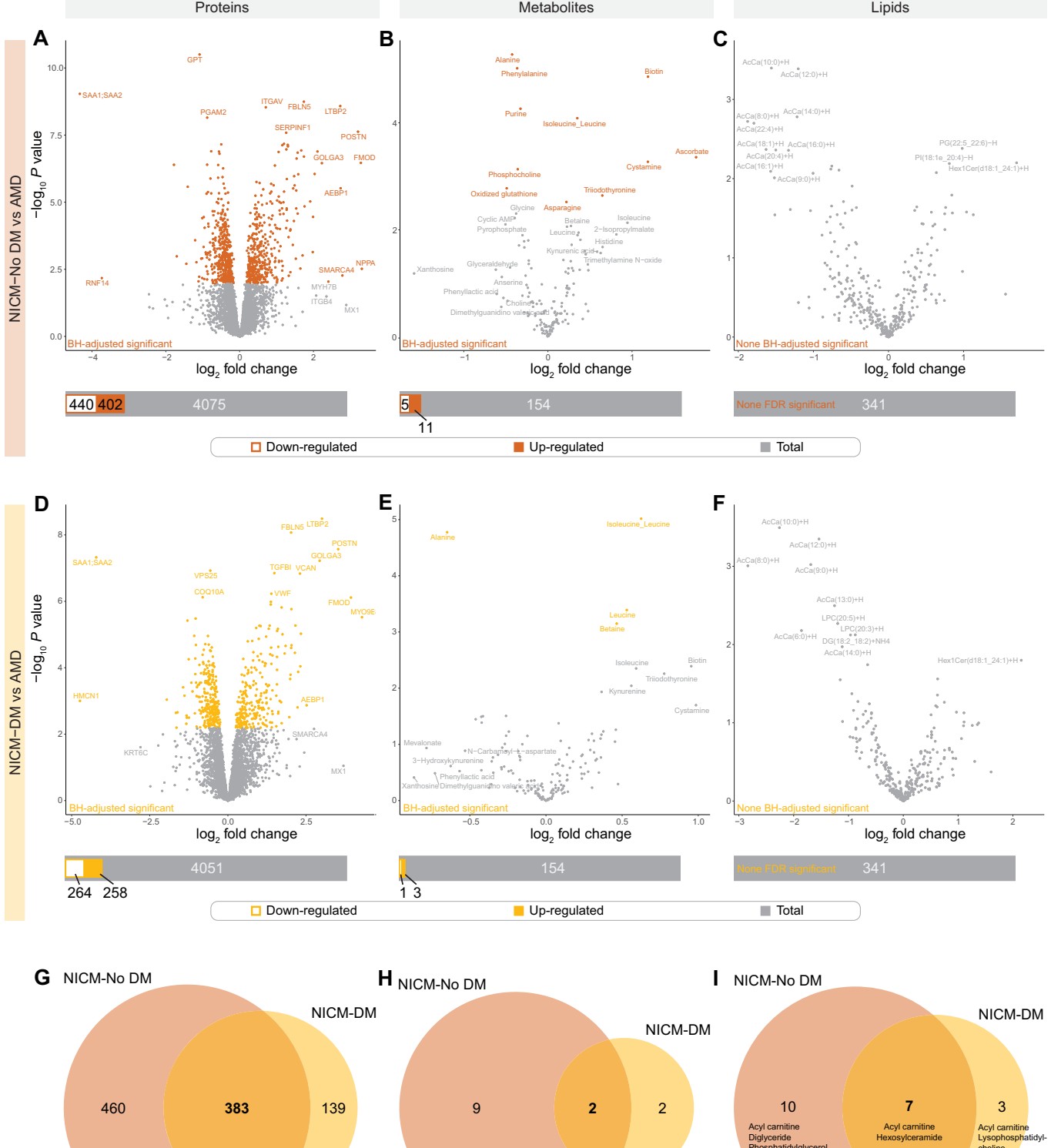

**Figure EV2. Human left ventricular myocardial differential analysis in protein, metabolite, and lipid abundance between non-ischaemic cardiomyopathy with (NICM-DM) and without diabetes (NICM-No DM) and age-matched donors (AMD).**

(A–F) Differential abundance was determined following Benjamini–Hochberg false discovery rate adjustment (FDR) of P values (FDR < 0.05). Analyses were performed using a moderated t test with the limma package (version 3.56.2) in R (version 4.3.1) following $\log_2$ transformation. NICM-DM $n = 9$, NICM-No DM $n = 18$ (proteomics and metabolomics) and 17 (lipidomics), AMD $n = 20$ (proteomics and metabolomics) and 19 (lipidomics). Superimposed bar plots summarise the number of significantly downregulated (white-filled bar) and upregulated (colour-filled bar) molecules relative to the total number of molecules analysed (grey bar). (A–C) NICM-No DM vs AMD. (D–F) NICM-DM vs AMD. (G, H) NICM-DM and NICM-No DM vs AMD FDR significant differentially abundant proteins and metabolites. (I) NICM-DM vs AMD and NICM-No DM vs AMD unadjusted significant (P < 0.05) lipids which were also > ±2-FC (in either direction) for comparative purposes (significance not accepted). Lipid classes annotated in order of the number of lipids which were unadjusted significant.

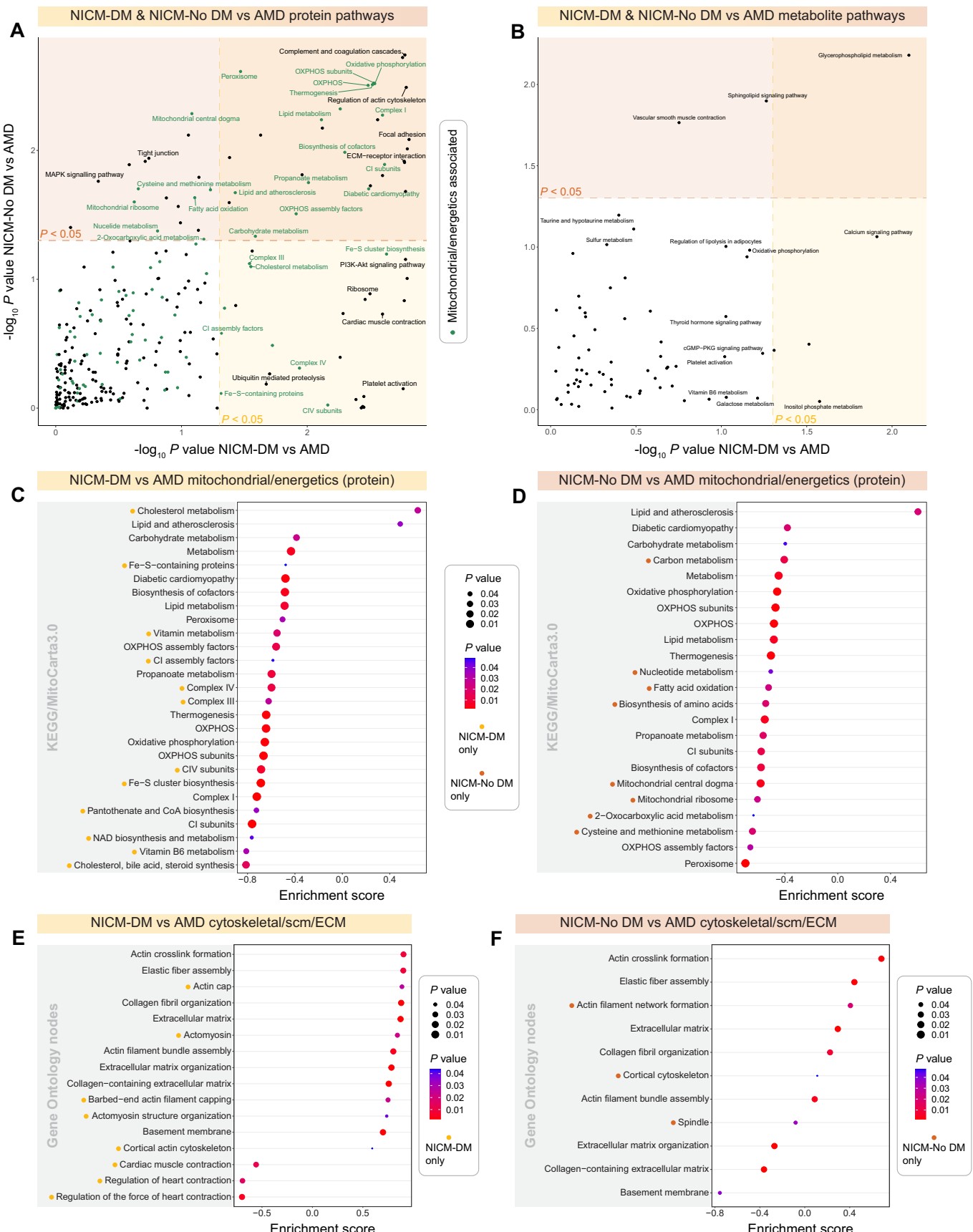

◀ **Figure EV3. Human non-ischaemic cardiomyopathy with (NICM-DM) and without diabetes (NICM-No DM) vs age-matched donor (AMD) myocardial protein and metabolite pathway analyses.**

(A, B) Scatter plots of -$\log_{10}$ P value enriched KEGG and MitoCarta3.0 protein pathways/gene sets of NICM-DM AMD and NICM-No DM where mitochondrial and energetics related pathways are coloured in green and significant ($P < 0.05$). Pathways in overlapping coloured regions were significant in both NICM-DM and NICM-No DM vs AMD. (A), Enriched pathways from the proteomic mass spectrometry (MS) analysis. (B) Enriched pathways from the metabolomic MS analysis. (C–F) Gene Set Enrichment Analysis (GSEA) enrichment bubble plots of top-ranked significantly enriched ($P < 0.05$) KEGG/MitoCarta3.0 pathways/gene sets and Gene Ontology nodes, respectively. (C, D) From (A). (E, F) Cytoskeletal, sarcomeric (scm), and extracellular matrix (ECM) enriched Gene Ontology Biological Process and Cellular Component nodes from selected parent nodes. Gene Set Enrichment Analysis (GSEA) analyses for (A–F) were performed on normalised and transformed proteomic and metabolomic datasets using clusterProfiler (version 4.8.1).

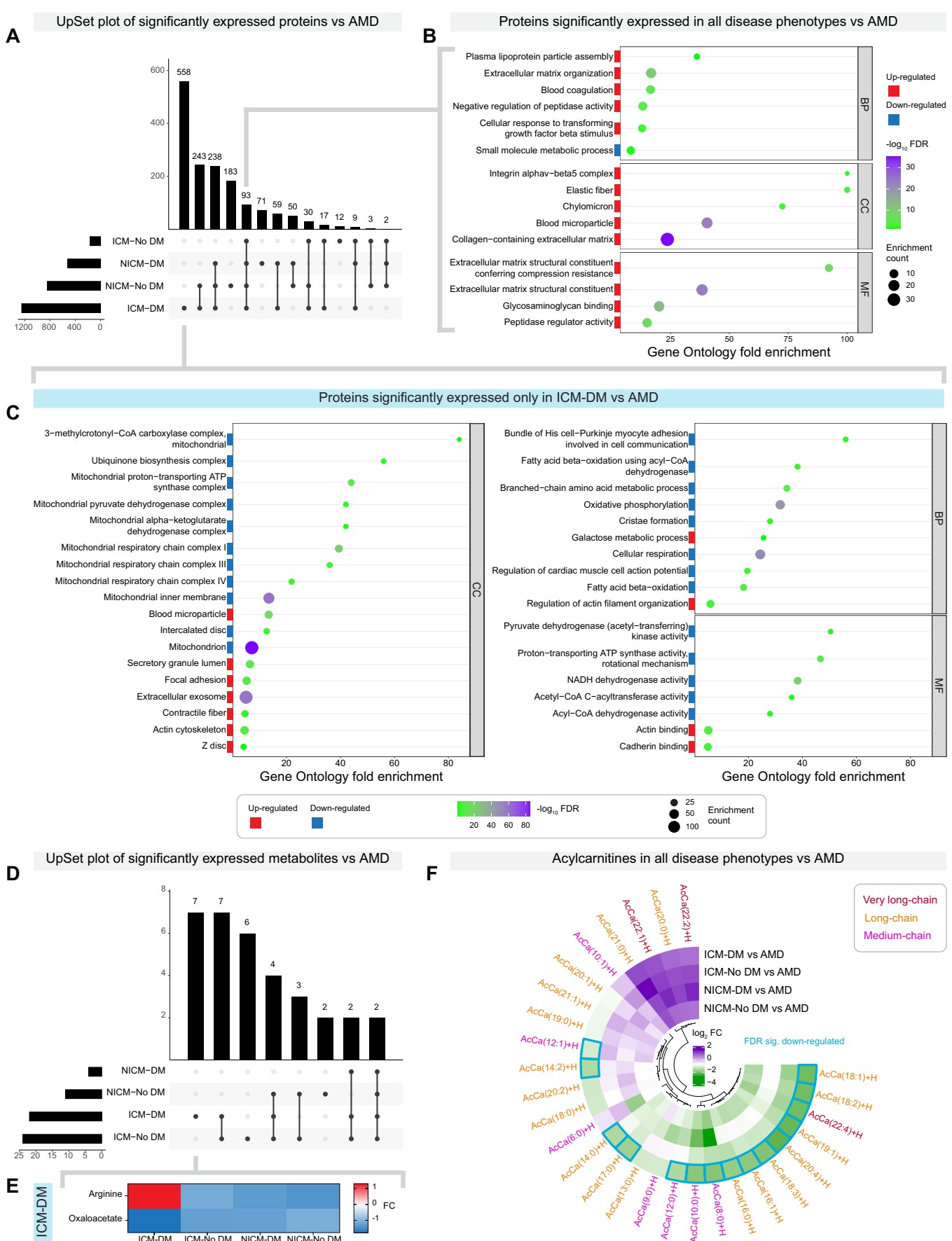

◀ **Figure EV4. Myocardial differential abundance co-analyses of all heart failure conditions vs age-matched donors (AMD), relative to each other, for proteomic, metabolomic, and lipidomic mass spectrometry analyses, with a focus on ischaemic cardiomyopathy with diabetes (ICM-DM).**

(**A**) UpSet plot summarising significant differentially expressed proteins in all the heart failure conditions; ICM-DM, ICM without diabetes (ICM-No DM), non-ischaemic cardiomyopathy with diabetes (NICM-DM), and NICM without diabetes (NICM-No DM), vs AMD from proteomic mass spectrometry (MS). Statistical significance of differential expression was determined following Benjamini–Hochberg false discovery rate adjustment (FDR) of $P$ values (FDR < 0.05). (**B**) Gene Ontology (GO) analysis by PANTHER (http://geneontology.org/, PANTHER17.0) enrichment bubble plot showing selected significantly enriched Biological Process (BP), Cellular Component (CC), and Molecular Function (MF) nodes from proteins which were significantly differentially expressed in all heart failure conditions vs AMD. Nodes with an FDR < 0.05 were considered statistically significant. This plot was the combination of two separate GO analyses; one from downregulated proteins compared to AMD (blue) and one from upregulated proteins compared to AMD (red). Enrichment count, represented as the size of the bubble, is the number of significant proteins in that particular node. GO fold enrichment is calculated as observed enrichment count/expected enrichment count from a random set of gene symbols of equal an input set size. (**C**) GO analysis by PANTHER enrichment bubble plot of selected significantly enriched nodes from proteins which were significantly expressed only in ICM-DM vs AMD. (**D**) UpSet plot summarising FDR significant differentially abundant metabolites in all the heart failure conditions vs AMD from metabolomic MS. (**E**) Heat map of the two greatest fold change (FC) metabolites in, and specific to, ICM-DM vs AMD. (**F**) Circular heatmap of all heart failure conditions $\log_2$ FC vs AMD quantified acylcarnitines from lipidomic MS where FDR significant differentially abundant lipids are indicated. A Lipid's chain length was contrasted to others in colour; very-long-chain (22 or more carbon length, maroon), long-chain (13–21 carbon length, orange), and medium-chain (6–12 carbon length, pink). Short-chain acylcarnitines (2–5 carbon length) were quantified in metabolomic MS.

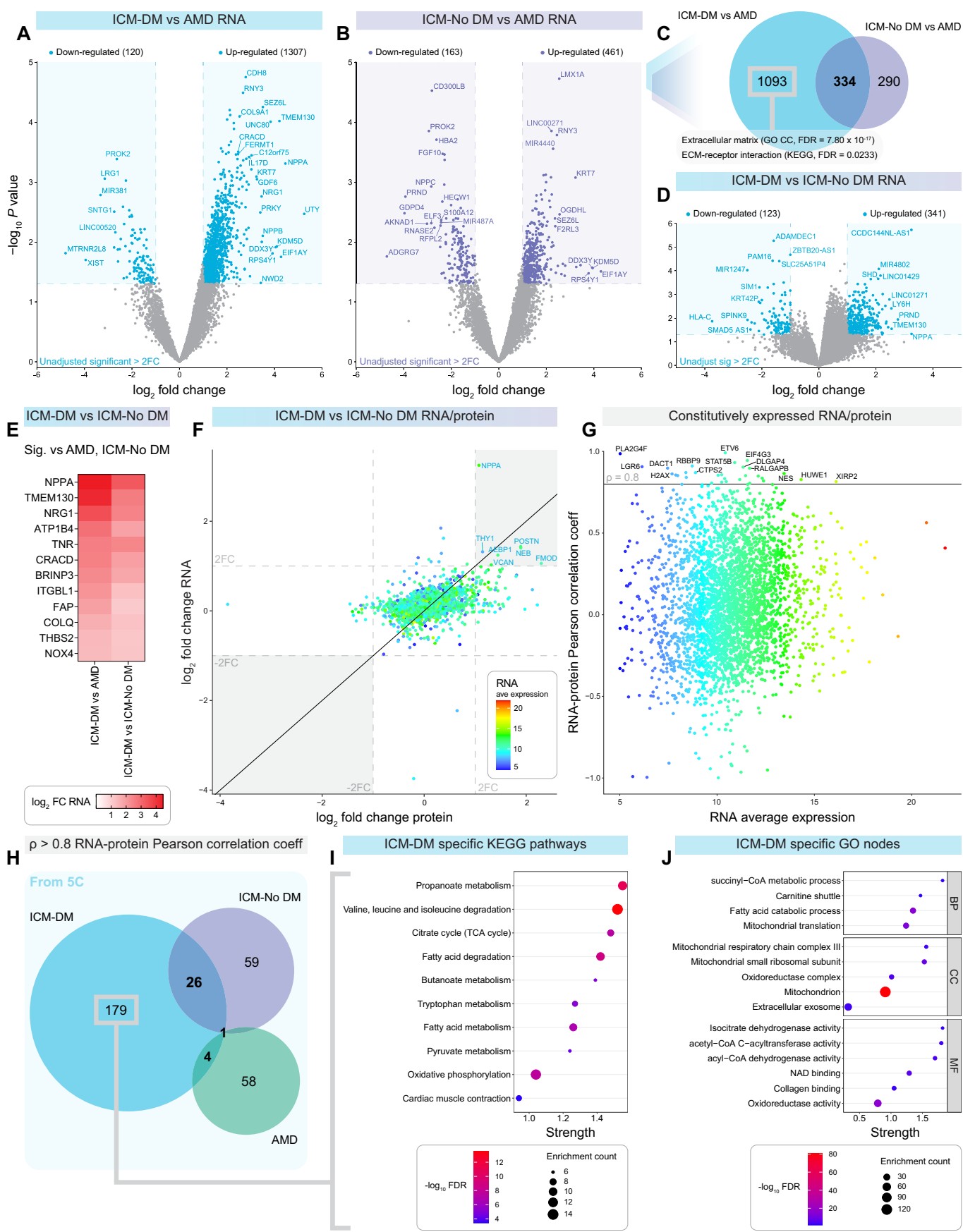

**Figure EV5.   Human left ventricular myocardium RNA sequencing analyses with protein co-expression and correlation in ischaemic cardiomyopathy (ICM) and age-matched donors (AMD).**

(A, B) Differential statistical significance was determined as unadjusted significant ($P = 0.05$) and > ±2-fold change (FC). Higher FC significant sequences, written as gene symbols, were annotated. (A), ICM with diabetes (ICM-DM) vs AMD. (B) ICM without diabetes (ICM-No DM) vs AMD. (C) Summary Venn diagram of (A, B). Sequences which were only significant in ICM-DM were analysed in STRING (https://string-db.org/, version 12.0) which revealed that the Extracellular matrix Gene Ontology (GO) Cellular Component (CC) node and the ECM-receptor interaction KEGG pathway were among the highest significantly enriched nodes/pathways. Nodes/pathways with a Benjamini–Hochberg false discovery rate adjusted $P$ value (FDR) < 0.05 were determined as significant. (D) ICM-DM vs ICM-No DM RNA. Down and upregulation was defined as significance in ICM-DM compared to ICM-No DM. (E) Summary heat map of RNA sequences of biological interest from (D) which were both significant in ICM-DM vs AMD and ICM-No DM. (F) RNA and protein $\log_2$ FC plot of ICM-DM vs ICM-No DM where gene symbols which are >2-FC in both protein and RNA are annotated. ICM-DM average RNA expression represented on a colour scale. (G) RNA–protein Pearson correlation plot of ICM-DM, ICM-No DM, and AMD combined to identify common constitutively expressed RNA/protein whereby a correlation coefficient of $\rho > 0.8$ was accepted as being correlated. Highest correlated gene symbols were annotated. Gene symbols were spread out across the x-axis according to average RNA expression of all the groups combined. (H) From (C), showing gene symbols which had an RNA–protein Pearson correlation coefficient of $\rho > 0.8$ in ICM-DM, ICM-No DM and AMD. (I, J) STRING (https://version-11-5.string-db.org/, version 11.5) analysis enrichment bubble plots of selected FDR significant (FDR < 0.05) KEGG pathways and Gene Ontology (GO) nodes from gene symbols which were RNA–protein correlated only in ICM-DM. Strength of the enriched pathways/nodes were calculated as $\log_{10}$(observed enrichment count/expected enrichment count from a random set of gene symbols of equal an input set size) wherein the enrichment count (represented as the size of the bubble) was the number of significant differentially expressed gene symbols in that particular node. (I) Enriched KEGG pathways. (J) Enriched GO Biological Process (BP), CC, and Molecular Function (MF) nodes. Differential analyses for (A, B, D) were performed using a moderated $t$ test with the limma package (version 3.56.2) in R (version 4.3.1) following $\log_2$ transformation. ICM-DM $n = 7$, ICM-No DM $n = 7$, AMD $n = 7$.

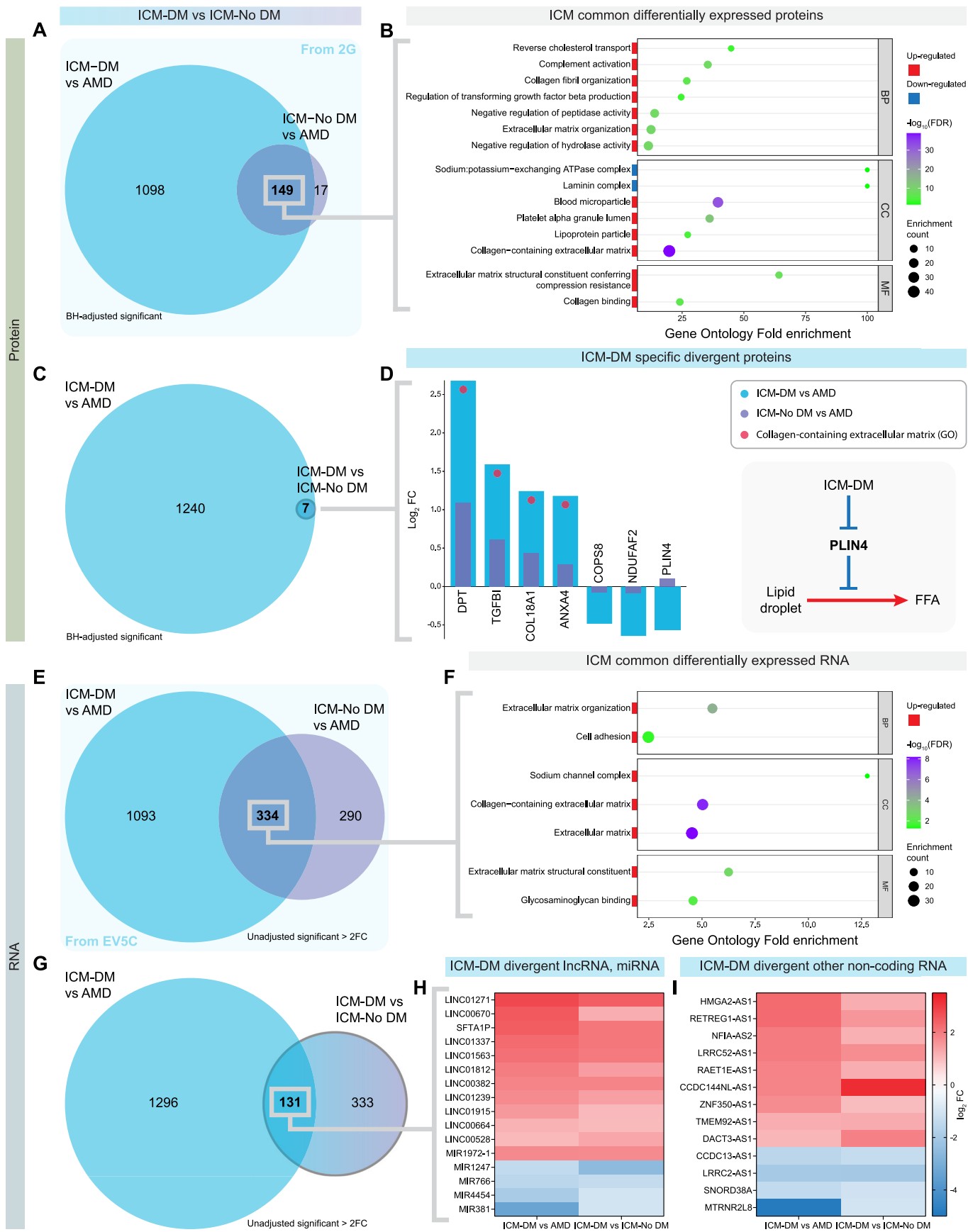

◀ **Figure EV6. Human ischaemic cardiomyopathy (ICM) common differentially expressed myocardial proteins and RNA, and ICM with diabetes (ICM-DM) specific divergent proteins and RNA.**

(A) From Fig. 2G of ICM-DM and ICM without diabetes (ICM-No DM) vs AMD significant differentially expressed proteins. Statistical significance was determined following Benjamini–Hochberg false discovery rate adjustment (FDR) of *P* values (FDR < 0.05). (B) Gene Ontology (GO) analysis by PANTHER (http://geneontology.org/, PANTHER17.0) enrichment bubble plot showing selected significantly enriched Biological Process (BP), Cellular Component (CC), and Molecular Function (MF) nodes from proteins which were significantly differentially expressed in both ICM-DM and ICM-No DM vs AMD. Nodes with an FDR < 0.05 were considered statistically significant. This plot was the combination of two separate GO analyses; one from downregulated proteins compared to AMD (blue) and one from upregulated proteins compared to AMD (red). Enrichment count, represented as the size of the bubble, is the number of significant proteins in that particular node. GO fold enrichment is calculated as observed enrichment count/expected enrichment count from a random set of gene symbols of equal an input set size. (C) ICM-DM vs AMD and ICM-DM vs ICM-No DM. (D) A log$_2$ fold change (FC) bar plot of ICM-DM and ICM-No DM vs AMD of proteins found to be significant in ICM-DM vs ICM-No DM (ICM-DM-specific and divergent from ICM-No DM). PLIN4, which was reduced in ICM-DM vs AMD while increased in ICM-No DM vs AMD, negatively regulates free fatty acid (FFA) availability. (E) Summary Venn diagram from Fig. EV5C of ICM-DM and ICM-No DM vs AMD significant differentially expressed RNA. Significance of RNA expression was determined as unadjusted significant (*P* = 0.05) and ±2-FC. (F) GO analysis by PANTHER of significant differentially expressed RNA in both ICM-DM and ICM-No DM vs AMD. (G) Venn diagram of significantly expressed RNA in ICM-DM vs AMD and ICM-DM vs ICM-No DM. (H) Long noncoding and microRNA sequences, and (I) other noncoding RNA significant in ICM-DM vs AMD and ICM-DM vs ICM-No DM in the same direction. Scale bar is shared between (H) and (I).

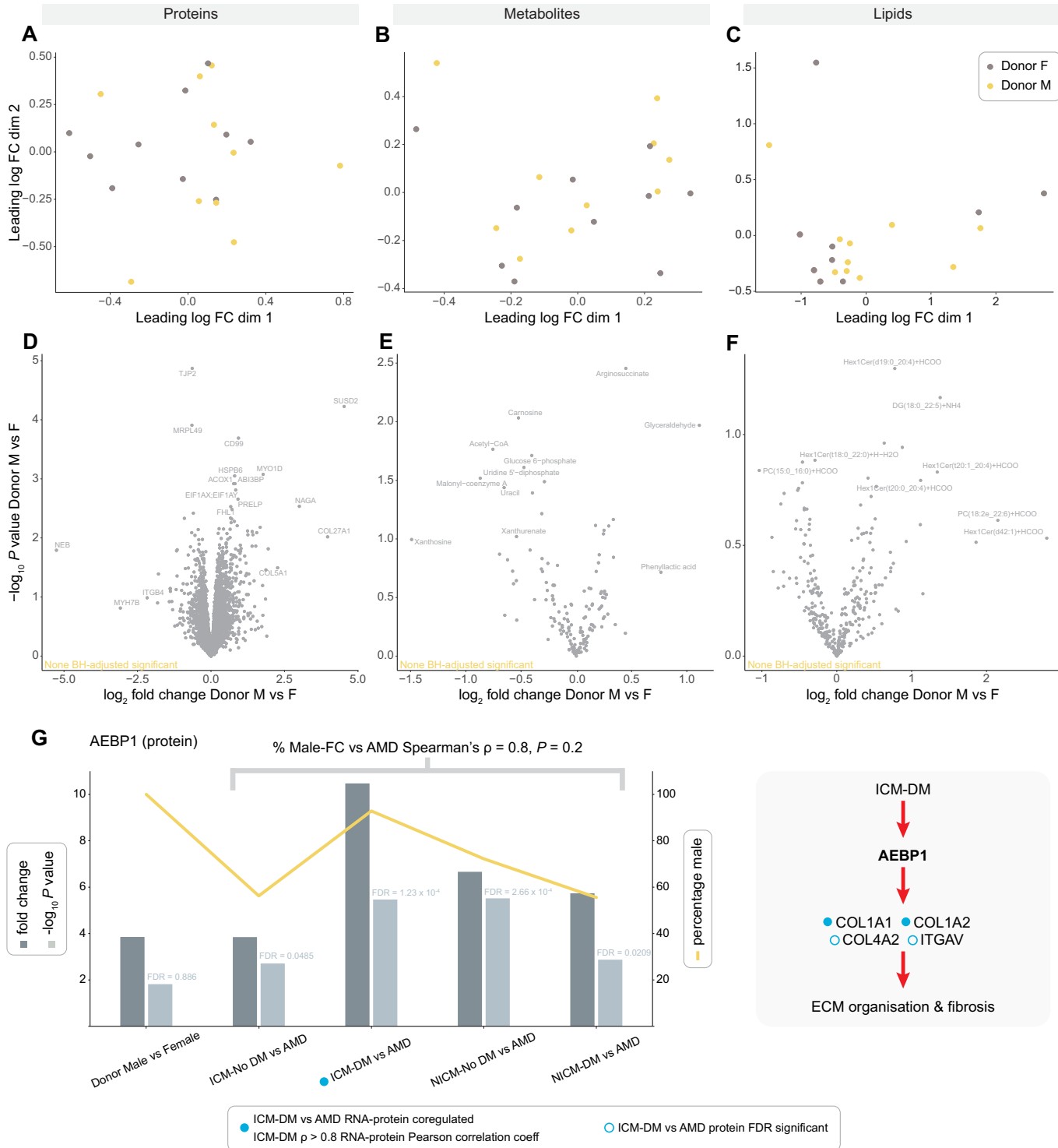

◀ **Figure EV7. Healthy donor male myocardium vs female and potential male-dominant influence in heart failure.**

(A–C) Multidimensional scaling (MDS) plots, a tool to visualise high dimensional data in two dimensions, were used to iteratively separate samples (each point) in distance based on their dissimilarity of paired data (proteins, metabolites, and lipids; dimensions) calculated as the leading $\log_2$ fold change (FC, average root-mean-square of the largest $\log_2$ FCs). Healthy donor males represented as yellow, females as grey. (D–F) Volcano plots of male vs female molecule (proteins, metabolites, and lipids, respectively) abundance whereby significance was determined after Benjamini–Hochberg false discovery rate adjustment (FDR) of $P$ values (FDR < 0.05). The was no significance in abundance between donor males and females. Analyses were performed using a moderated $t$ test with the limma package (version 3.56.2) in R (version 4.3.1) following $\log_2$ transformation. AMD males $n = 10$, AMD females $n = 10$ (proteomics and metabolomics) and 9 (lipidomics). (G) Bar plot of protein AEBP1 fold change (dark grey) and $\log_{10}$ $P$ value (light grey, with FDR-adjusted value annotated) in donor males vs females, and heart failure conditions vs age-matched donors (AMD). Superimposed line plot shows the associated percentage of males in the subject group; the donor male group are 100% male. AEBP1 was FDR significant differentially expressed in all HF conditions vs AMD but the most in ICM-DM, which also had the highest FC. The ICM-DM group had the highest percentage of males. A male-protein FC vs AMD nonparametric Spearman's correlation test was performed on the HF conditions to determine a possible significant male relationship to AEBP1 FC. Significance was determined as $\rho > 0.8$ and $P < 0.05$. AEBP1 has been identified to positively regulate extracellular matrix (ECM) organisation and fibrosis, and was co-regulated in ICM-DM vs AMD in both RNA and protein, and RNA–protein correlated in ICM-DM.

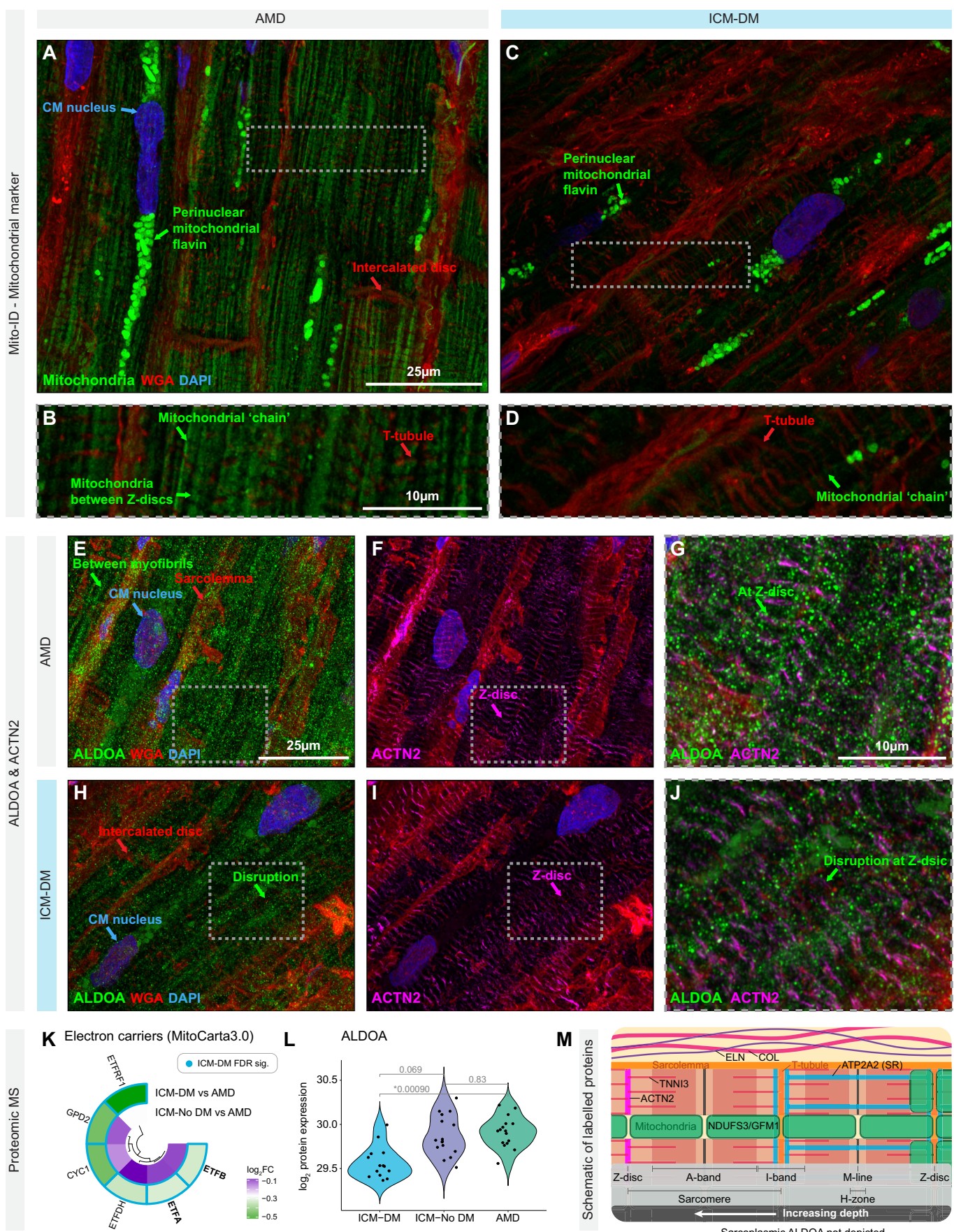

**Figure EV8. Immunofluorescent labelling of mitochondria and sarcoplasmic ALDOA in cryopreserved ischaemic cardiomyopathy with diabetes (ICM-DM) and healthy age-matched donor (AMD) myocardium.**

(A–D) Immunofluorescent confocal microscopy images of cryopreserved human myocardium representing qualitative differences between AMD and ICM-DM in mitochondria labelled with Mito-ID. Bright (autofluorescent) perinuclear mitochondrial flavin identified. (E–J) Labelled muscle aldolase (ALDOA, green), localised in the sarcoplasm, with a labelled Z-disc protein, ACTN2 (magenta). Membranes, particularly the sarcolemma, were stained with fluorophore-conjugated wheat germ agglutinin (red) and nuclei were stained using DAPI (blue). Cardiomyocytes (CM) are depicted in a longitudinal orientation. All images are 4.5 μm-thick Z-stacks, deconvolved using Huygens Professional, and compressed into a two-dimensional image using Fiji/ImageJ Maximum Intensity Projections. (K) Proteomic mass spectrometry (MS) $log_2$FC circular heatmap of quantified proteins from the MitoCarta3.0 electron carriers gene set. (L) Violin plot of proteomic MS $log_2$ transformed quantification of ALDOA in ICM-DM, ICM without diabetes (ICM-No DM), and AMD groups. FDR-adjusted $P$ values were annotated to reveal significant differences (*, FDR < 0.05) between groups. Analyses were performed using a moderated $t$ test with the limma package (version 3.56.2) in R (version 4.3.1) following $log_2$ transformation. ICM-DM $n = 14$, ICM-No DM $n = 16$, AMD $n = 20$. (M) Cellular schematic showing localisation of all histologically labelled proteins in this study. Sarcoplasmic ALDOA not depicted. SR sarcoplasmic reticulum. All tissue was pre-mortem. Macroscopic scar tissue, particularly in heart failure conditions, was avoided in all quantitative analyses (mass spectrometry and RNA sequencing) and imaging. Only the most normal/healthy appearing and longitudinally oriented AMD and ICM-DM cardiomyocytes were imaged in confocal microscopy.

