## [Peer Review File · EMBO Molecular Medicine]

Left ventricular myocardial molecular profile of human diabetic ischaemic cardiomyopathy

Benjamin Hunter, Yunwei Zhang, Dylan Harney, Holly McEwen, Yen Koay, Michael Pan, Cassandra Malecki, Jasmine Khor, Robert Hume, Giovanni Guglielmi, Alicia Walker, Shashwati Dutta, Vijay Rajagopal, Anthony Don, Mark Larance, John O'Sullivan, Jean Yee Hwa Yang, and Sean Lal

Corresponding author: Sean Lal (sean.lal@sydney.edu.au)

Review Timeline:

Submission Date:	26th Feb 25
Editorial Decision:	31st Mar 25
Revision Received:	10th Jun 25
Editorial Decision:	26th Jun 25
Revision Received:	11th Jul 25
Accepted:	15th Jul 25

Editor: Lise Roth

Transaction Report:

31st Mar 2025

Dear Prof. Lal,

Thank you for submitting your manuscript to EMBO Molecular Medicine and please accept my apologies for the delay in getting back to you as we were waiting for a referee report. Unfortunately, referee #1 has not yet got back to us, but given that both referees #2 and #3 make similar recommendations, we prefer to make a decision now to avoid further delay in the process. Should referee #1 provide a report, we will send it to you on the understanding that we will not ask you to carry out any extensive experiments beyond those requested in the attached reports from referees #2 and #3. As you will see from the reports below, the referees recognise the interest of the study and generally support publication of your work, subject to appropriate revisions.

Addressing the reviewers' concerns in full will be necessary for further considering the manuscript in our journal, and acceptance of the manuscript will entail a second round of review. However, as the manuscript is a resource, please note that we do NOT ask for functional validation of your findings. EMBO Molecular Medicine encourages a single round of revision only and therefore, acceptance or rejection of the manuscript will depend on the completeness of your responses included in the next, final version of the manuscript. For this reason, and to save you frustration at the end, I would strongly discourage you from returning an incomplete revision.

We are expecting your revised manuscript within three to four months, if you anticipate any delay, please contact us.

We require:

4) A .docx formatted letter INCLUDING the reviewers' reports and your detailed point-by-point responses to their comments. As part of the EMBO Press transparent editorial process, the point-by-point response is part of the Review Process File (RPF), which will be published alongside your paper.

5) A complete author checklist, which you can download from our author guidelines (<https://www.embopress.org/page/journal/17574684/authorguide#submissionofrevisions>). Please insert information in the checklist that is also reflected in the manuscript. The completed author checklist will also be part of the RPF.

6) All Materials and Methods need to be described in the main text using our 'Structured Methods' format. According to this format, the Methods section includes a Reagents and Tools Table (listing key reagents, experimental models, software and relevant equipment and including their sources and relevant identifiers) followed by a Methods and Protocols section describing the methods, ideally using a step-by-step protocol format. The aim is to facilitate adoption of the methodologies across labs. Please download and fill our Reagents and Tools Table template (.docx), which you can find in our author guidelines: <https://www.embopress.org/page/journal/14693178/authorguide#structuredmethods>.

<https://www.embopress.org/doi/10.15252/msb.20178071>

7) Please note that all corresponding authors are required to supply an ORCID ID for their name upon submission of a revised manuscript.

8) It is mandatory to include a 'Data Availability' section after the Materials and Methods. Before submitting your revision, primary datasets produced in this study need to be deposited in an appropriate public database, and the accession numbers and database listed under 'Data Availability'. Please remember to provide a reviewer password if the datasets are not yet public (see <https://www.embopress.org/page/journal/17574684/authorguide#dataavailability>).

9) For data quantification: please specify the name of the statistical test used to generate error bars and P values, the number (n) of independent experiments (specify technical or biological replicates) underlying each data point and the test used to calculate p-values in each figure legend. The figure legends should contain a basic description of n, P and the test applied. Graphs must include a description of the bars and the error bars (s.d., s.e.m.). Please provide exact p values.

10) Our journal encourages inclusion of *data citations in the reference list* to directly cite datasets that were re-used and obtained from public databases. Data citations in the article text are distinct from normal bibliographical citations and should directly link to the database records from which the data can be accessed. In the main text, data citations are formatted as follows: "Data ref: Smith et al, 2001" or "Data ref: NCBI Sequence Read Archive PRJNA342805, 2017". In the Reference list, data citations must be labeled with "[DATASET]". A data reference must provide the database name, accession number/identifiers and a resolvable link to the landing page from which the data can be accessed at the end of the reference. Further instructions are available at .

11) We replaced Supplementary Information with Expanded View (EV) Figures and Tables that are collapsible/expandable online. EV Figures should be cited as 'Figure EV1, Figure EV2' etc... in the text and their respective legends should be included in the main text after the legends of regular figures.

12) The paper explained: EMBO Molecular Medicine articles are accompanied by a summary of the articles to emphasize the major findings in the paper and their medical implications for the non-specialist reader. Please provide a draft summary of your article highlighting

13) Author contributions: CRedit has replaced the traditional author contributions section because it offers a systematic machine readable author contributions format that allows for more effective research assessment. Please remove the Authors Contributions from the manuscript and use the free text boxes beneath each contributing author's name in our system to add specific details on the author's contribution. More information is available in our guide to authors.

Please also suggest a visual abstract to illustrate your article as a PNG file 550 px wide x 300-600 px high. A cropped portion of this image will serve as thumbnail for the table of content on our webpage.

16) As part of the EMBO Publications transparent editorial process initiative (see our Editorial at <http://embomolmed.embopress.org/content/2/9/329>), EMBO Molecular Medicine will publish online a Review Process File (RPF) to accompany accepted manuscripts.

In the event of acceptance, this file will be published in conjunction with your paper and will include the anonymous referee reports, your point-by-point response and all pertinent correspondence relating to the manuscript. Let us know whether you agree with the publication of the RPF and as here, if you want to remove or not any figures from it prior to publication. Please note that the Authors checklist will be published at the end of the RPF.

I look forward to receiving your revised manuscript.

Yours sincerely,

Lise Roth

**** Reviewer's comments ****

Referee #2 (Comments on Novelty/Model System for Author):

Given the difficulty in collecting these clinical samples (particularly for those non-ischaemic cardiomyopathy with diabetes), findings from these omics studies will provide valuable information for the understanding of pathophysiological changes of diabetic ischemic cardiomyopathy. However, there is no functional validation of any of the findings in this submission, making the findings all association and the conclusion suggestive.

Referee #2 (Remarks for Author):

This study uses multi-omics (proteomic, metabolomic, lipidomic, and RNAseq) to profile left ventricular myocardium from patients with ischaemic and non-ischaemic cardiomyopathy with and without diabetes (vs healthy age-matched donors). Given the difficulty in collecting these clinical samples (particularly for those non-ischaemic cardiomyopathy with diabetes), findings from these omics studies will provide valuable information for the understanding of pathophysiological changes of diabetic ischemic cardiomyopathy. The use of mitochondria and oxidative phosphorylation simulated modelling adds in-depth insight for the analysis. However, given all of these, there are concerns need to be addressed:

1. The whole studies are all about multi-omics and analysis. Without any functional validation in cells or in vivo in mice, the findings would be all association and conclusion would be suggestive. Functional validation studies with interventions of the key pathways/molecules identified from the omics analysis, will add a lot more value on top of these analysis.
2. I go through all the files couple times, but I can not find supplemental tables, not sure how the patients demographic data look like, for example, whether there will be big difference in anti-diabetic drugs used among groups.
3. It is somewhat not surprising to see those differences in mitochondrial composition, oxidative phosphorylation among different groups. So then the question would be rather than as a valuable resource this multi-omics studies provides, what new information generated from this study.
4. The pair analysis of RNA sequencing and omics data is interesting. If this could also integrate public data on whether a protein is secreted protein or not, may provide a better understanding the sources of these proteins and the potential way they function (e.g., inter-organ communication).
5. The histochemistry experiment is interesting. Does the area of the staining is the same area of samples subjected to multi-omics analysis?
6. For a resource purpose, a short report format would be more appropriate for this.

Referee #3 (Comments on Novelty/Model System for Author):

The authors produced a large amount of data that could be used to differentiate between the different manifestations of heart disease {plus minus} diabetes. The medical impact is medium because they do not provide a clear molecular signature to assess the disease in patients.

Referee #3 (Remarks for Author):

The authors performed a very comprehensive multi-omic analysis of post-mortem myocardial biopsies from IDM{plus minus}DM, NIDM{plus minus}DM patients and aged-matched donors (AMD). They identified a number of up- and down-regulated proteins associated with HF and proposed potential myocardial mitochondrial protein biomarkers. They also provide a link to a website with their data.

Comments:

1. The authors focus on numerous proteins that differ between DM vs. non-DM patients within the ICM and NICM cohort. Can they delineate a molecular signature to identify each disease subtype?
2. Did the authors observe any sex differences within disease groups (or AMD) in their analyses?
3. In Suppl Fig. 10, panels b and d are blurry and pixelated. They appear to be zoomed-in images of the delineated region in panels a and c rather than a higher magnification micrograph.

Reviewer's commentsAuthor responses in blue

Referee #2 (Comments on Novelty/Model System for Author):

Given the difficulty in collecting these clinical samples (particularly for those non-ischaemic cardiomyopathy with diabetes), findings from these omics studies will provide valuable information for the understanding of pathophysiological changes of diabetic ischemic cardiomyopathy. However, there is no functional validation of any of the findings in this submission, making the findings all association and the conclusion suggestive.

As noted by the Editors, this is a pure resource paper. We hope that functional studies will follow by others who use our study as a resource.

Referee #2 (Remarks for Author):

This study uses multi-omics (proteomic, metabolomic, lipidomic, and RNAseq) to profile left ventricular myocardium from patients with ischaemic and non-ischaemic cardiomyopathy with and without diabetes (vs healthy age-matched donors). Given the difficulty in collecting these clinical samples (particularly for those non-ischaemic cardiomyopathy with diabetes), findings from these omics studies will provide valuable information for the understanding of pathophysiological changes of diabetic ischemic cardiomyopathy. The use of mitochondria and oxidative phosphorylating simulated modelling adds in-depth insight for the analysis. However, given all of these, there are concerns need to be addressed:

1. The whole studies are all about multi-omics and analysis. Without any functional validation in cells or in vivo in mice, the findings would be all association and conclusion would be suggestive. Functional validation studies with interventions of the key pathways/molecules identified from the omics analysis, will add a lot more value on top of these analysis.

We thank the Reviewer for their comment. As noted by the Editors, this is a pure resource paper. We hope that functional studies will follow by others who use our study as a resource.

Though there are no functional investigations in this study, we believe the volume and breadth of the human samples, and the quantitative and qualitative techniques used justify it as a high-value resource for future research.

2. I go through all the files couple times, but I can not find supplemental tables, not sure how the patients demographic data look like, for example, whether there will be big difference in anti-diabetic drugs used among groups.

We thank the Reviewer for their observation and attention to detail. The diabetic patients were using Metformin and insulin. There were no significant differences in their diabetic care. The long-standing diabetes had been a contributor to their ischaemic heart failure. This information has been added to '*Acquisition of human myocardium*' in Materials and Methods of the manuscript.

3. It is somewhat not surprising to see those differences in mitochondrial composition, oxidative phosphorylation among different groups. So then the question would be rather than as a valuable resource this multi-omics studies provides, what new information generated from this study.

We thank the Reviewer for their comment and appreciate the importance of highlighting new findings. Though this manuscript is described as a resource paper, it is not simply a resource paper in a strict sense.

The Results first identifies condition similarity (Figures EV1 and EV9A-C) and comprehensively presents all differentially abundant molecules in the heart failure conditions compared to healthy age-matched donors (Figures 2, EV2, EV3, EV7A-D, and EV9D-F), then focuses on isolating specific heart failure conditions and components of them, including mitochondrial proteins and products (Figures 3, 4, EV4), and removing confounding variables (Figures EV6C-D and EV9). Where a pure resource paper may end there, in later part of the Results and in the Discussion, all the histochemistry (Figures 6, EV10, and EV11) and multi-omics results (protein, RNA, metabolites, and lipids) are integrated into intracellular pathways and mechanisms (Figures 5, EV7F-J, and EV8) linking previous findings from humans and animals, and inferring meaning and identifying real and new foundational changes induced by diabetes in ischaemic cardiomyopathy.

To our knowledge, this is the first study to interrogate the molecular profile of the same cohort of diseased left ventricular myocardium using four different multi-omics techniques and across so many pre-mortem heart failure conditions. In multi-omics, merging separate datasets from different studies is highly unfavorable as the equipment, methods and protocol, and the acquisition and quantification of molecules can vary greatly. Herein, this study captures a great breadth of information which would be hard to replicate from collating data from previous studies. Therefore, all the Discussion presents new information, taken as a whole, verified by previous studies. Some of these key findings are summarized in the Synopsis (and below).

We appreciate the Reviewer's point and agree that mitochondrial dysfunction has been previously associated with heart failure and diabetes. Myocardial maladaptive remodeling is also well documented in heart failure. However, our study provides several new heart failure-specific insights that, to our knowledge, have not been previously described:

- We show that mitochondrial complex I (CI) subunit down-regulation is significantly more pronounced in ICM-DM than in other forms of HF, with 90 OXPHOS genes differentially abundant in ICM-DM compared to 27–35 in others.
- We demonstrate that medium to very long-chain acylcarnitines are uniquely down-regulated in ICM-DM, indicating a distinct lipid metabolic impairment not seen in other HF conditions (Figures 2F and EV5F).
- Through paired RNA–protein analysis, we identify a subset of extracellular matrix-related genes (e.g., COL1A2, AEBP1) that are not only up-regulated at both transcript and protein levels but are co-regulated specifically in ICM-DM, suggesting transcriptional coordination of fibrosis pathways.
- We identify novel biomarkers of mitochondrial volume/activity (SUCLG1, CKMT2, FH, and ECHS1) via correlation with the entire mitochondrial proteome, which may serve as disease-specific indicators.
- We provide confocal microscopy validation of protein disorganisation (ATP2A2, ALDOA, and GFM1) along the sarcomere in ICM-DM myocardium, strengthening the biological relevance of our omic findings.

Taken together, our study goes beyond confirming known mitochondrial involvement and identifies a unique, diabetes-associated molecular profile in human ICM, offering specific candidates for future investigation.

4. The pair analysis of RNA sequencing and omics data is interesting. If this could also integrate public data on whether a protein is secreted protein or not, may provide a better understanding the sources of these proteins and the potential way they function (e.g., inter-organ communication).

We thank the Reviewer for their thoughtful question and insight. Indeed, this study collectively analyses proteins and RNA from the same cohort of individuals, and from the outer anterior or lateral walls of the left ventricular myocardium, without macroscopic scar tissue, to indicate if the changes in protein expression are done so at the transcriptional or post-translational level. The myocardium was cryogenically stored after perfusion, so the quantification of proteins and RNA would be almost entirely intrinsic to the myocardium.

We agree that identifying secreted proteins could inform understanding of protein origin and function. However, as the myocardial tissue was perfused prior to snap freezing, circulating or liver-derived plasma proteins are expected to be minimal. Therefore, we have generated

new Expanded View Tables (EV72 and EV73) featuring known extracellular matrix secretome proteins and transcripts from the Human Protein Atlas and how they were differentially expressed in our samples (Uhlén *et al*, 2019). While this provides insight into local remodelling and myocardial signalling, inter-organ communication cannot be directly inferred from this dataset due to the tissue preparation method.

The Reviewer poses an interesting question regarding serum proteins potentially influencing the heart from other locations. Shah *et al* in 2024 identified 37 serum proteins associated with heart failure independent of traditional risk factors (Shah *et al*, 2024). To satisfy the Reviewer's question, these 37 proteins were cross-referenced with the proteins analysed in our study, as well as the Human Protein Atlas secretome (Table EV 74). This revealed 12 of the 37 proteins to be significantly differentially abundant in ICM-DM, of which, 9 were secretory proteins (FBLN5, CLEC3B, MFAP4, SPON1, C9, NRP1, IGFBP7, APOF, ITIH3). Proteins FBLN5 and MFAP4 were secretory proteins of the extracellular matrix. However, some of these serum proteins may have originated from the heart. This can be found in the Discussion of the manuscript under '*Regulators of the extracellular matrix*'.

5. The histochemistry experiment is interesting. Does the area of the staining is the same area of samples subjected to multi-omics analysis?

We thank the Reviewer for their complement and question. Yes, the area of the staining is the same area of the same samples subjected to multi-omics analysis (i.e. outer anterior and lateral walls of the left ventricle). As described in the Methods, the quality of tissue used for the quantification and the histochemistry were equal and were without macroscopic scar tissue.

6. For a resource purpose, a short report format would be more appropriate for this.

We thank the Reviewer for their comment. The Editors are happy with the paper being a full-length resource paper. We believe the length of the paper is appropriate for the scale of the research and to adequately communicate interesting findings to the reader, as highlighted by Reviewer #3's points 1 and 2 below (signature molecules and a sex analysis).

Referee #3 (Comments on Novelty/Model System for Author):

The authors produced a large amount of data that could be used to differentiate between the different manifestations of heart disease {plus minus} diabetes. The medical impact is medium because they do not provide a clear molecular signature to assess the disease in patients.

We thank the Reviewer for their observation. As noted by the Editors, this is a pure resource paper. We hope that functional studies will follow by others who use our study as a resource.

Referee #3 (Remarks for Author):

The authors performed a very comprehensive multi-omic analysis of post-mortem myocardial biopsies from IDM{plus minus}DM, NIDM{plus minus}DM patients and aged-matched donors (AMD). They identified a number of up- and down-regulated proteins associated with HF and proposed potential myocardial mitochondrial protein biomarkers. They also provide a link to a website with their data.

Comments:

1. The authors focus on numerous proteins that differ between DM vs. non-DM patients within the ICM and NICM cohort. Can they delineate a molecular signature to identify each disease subtype?

We thank the Reviewer for their complement and question. As mentioned, our analyses shown in our Figures, Expanded View Figures, and our Expanded View Tables were comprehensive, and included the analysis of individual molecular profiles of each heart failure condition (Figures 2, 3, 4, 5, EV1, EV2, EV3, EV4, EV6, EV7, and EV8).

To satisfy the Reviewer's comment, we have added additional analyses in the Results of the manuscript to better highlight isolated and specific molecular profiles of ischaemic cardiomyopathy and diabetes; '*Ischaemic cardiomyopathy with diabetes and diabetes-specific targeted analyses*', and, '*Ischaemic cardiomyopathy with and without diabetes comparison*'.

To aide the reading audience, we have also identified signature proteins and metabolites specific to each heart failure condition by creating new Expanded View Tables, Tables EV65-71. These tables contain all the proteins and metabolites which were differentially abundant only in ICM-DM, ICM-No DM, NICM-DM or NICM-No DM. These tables expand from Figures EV5A and EV5D.

2. Did the authors observe any sex differences within disease groups (or AMD) in their analyses?

We thank the Reviewer for their question. Yes, sex analyses were performed and are represented in Figure EV9 and in Tables EV55-EV58. The analysis was mostly omitted in the manuscript text due to prioritizing other content within the word limit. Concisely written

results of the analysis have now been added in Results under the sub-heading '*Sex influence in multi-omic MS analyses*'.

3. In Suppl Fig. 10, panels b and d are blurry and pixelated. They appear to be zoomed-in images of the delineated region in panels a and c rather than a higher magnification micrograph.

We thank the Reviewer for their comment. The figures initially submitted were of lower resolution. The final submitted figures are now in individual files and are of higher resolution. Panels B and D, of which are insets of panels A and C, are zoomed-in and cropped versions of those images. Please note that the original images are very large (2048 x 2048) and there has been no digital zoom. There is no pixelation in the final images but a zoomed-in representation of the signal at a small scale across $\sim 40\mu\text{m}$.

References

- Shah AM, Myhre PL, Arthur V, Dorbala P, Rasheed H, Buckley LF, Claggett B, Liu G, Ma J, Nguyen NQ *et al* (2024) Large scale plasma proteomics identifies novel proteins and protein networks associated with heart failure development. *Nat Commun* 15: 528-528
- Uhlén M, Karlsson MJ, Hober A, Svensson A-S, Scheffel J, Kotel D, Zhong W, Tebani A, Strandberg L, Edfors F *et al* (2019) The human secretome. *Science signaling* 12

26th Jun 2025

Dear Prof. Lal,

Thank you for submitting your revised study to EMBO Molecular Medicine. As you will see from the reports below, referees #2 and #3 evaluated your revisions and are now satisfied. I will therefore be able to accept your manuscript once the following editorial concerns are addressed:

1/ Manuscript text:

- Please remove the yellow highlighted text, and only indicate in track changes mode any new modification in the text.
- There is a discrepancy between Jean Yang in the manuscript, vs. Jean Yee Hwa Yang in the submission system, please adjust.
- Please provide up to 5 keywords.
- Materials and Methods should be renamed Methods:
 - o Please remove the Reagent and Tools Table from the manuscript and upload it as a separate file choosing the file type "Reagent Table".
 - o Studies involving human participants: Include a statement confirming that the experiments conformed to the principles set out in the WMA Declaration of Helsinki and the Department of Health and Human Services Belmont Report.
 - o Statistics: please provide a statement on blinding and randomization.
 - o Please add a "Graphics" section to the Methods as follows:

Graphics:

(some of the... OR Figure #... OR synopsis) graphics were created with BioRender.com.

- Data Availability: Datasets should be publicly available before acceptance. When missing, please provide URLs. Code Availability should be merged with this section.
- Competing interests: please rename "Disclosure statement and competing interests".
- Funding: the following funders and grant numbers have been provided in the Comments box - they need to be removed from there and entered as separate funder each (via More Funders option): a Postdoctoral Research Fellowship from the School of Mathematics and Statistics, University of Melbourne. Metabolomics Workbench raw data availability is supported by NIH grant U2C-DK119886 and OT2-OD030544 grants

2/ Figures:

- Figure Callouts: panels A and B for Fig. 1 should be referenced in the manuscript text, and individual panels should also all be called out for the EV Figures; a callout is missing for Fig 2I.
- Please rename your Expanded View Tables "Dataset EV1". Please correct the callouts in the manuscript accordingly.
- As you currently have 6 EV figures and 11 EV figures, we would suggest merging some of the EV figures together, or including some of them in an Appendix file. The Appendix file is a PDF file and should start with a short Table of Content with page numbers. Appendix figures should be referred to in the main text as: "Appendix Figure S1, Appendix Figure S2" etc. The nomenclature of the EV figures in the figure legends should be corrected to Figure EV1, etc. instead of Figure Expanded View 1.
- Figure EV11 C-H: Please indicate in the legends whether the (paired) images are identical/consecutive sections.
- Please address the queries from our data editors:
 1. Please indicate the statistical test used for data analysis in the legends of figures 2A-F; 3A-F; 6M-P; EV2 A-F; EV3 A-F; EV4 A-F; EV7 A, B, D, I, J; EV9 A-E; EV10 L
 2. Please note that information related to n is missing in the legends of figures 2A-F; 5H-K, 6M-P; EV2 A-F; EV3 A-F; EV7 A, B, D; EV9 A-E; EV10 L
 3. Please note that for heatmap present in figures EV8 H, I a numbered scale bar is not provided. This needs to be rectified.

3/ Source Data: We note that most of your raw data is deposited in public repositories, however please also provide Source Data for the microscopy images in Figure 6.

4/ Checklist:

- top left corner, journal submitted: please indicate EMBO Molecular Medicine
- for all filled entries, please also fill in the right section, which indicates where in the manuscript the information is available.
- please fill in the complete 'Experimental study design and statistics' section.
- please fill in the 'Sample definition and in-laboratory replication' section.
- data availability: please check if you need to fill in the 'Computational models' subsection.

5/ Thank you for providing a synopsis image. Kindly note that the file needs to be jpeg, TIFF or png format, 550 px wide x 300-600 px high. I would suggest keeping only the top part of the image (study design). A cropped portion of this image will serve as thumbnail for the table of content on our webpage.

6/ As part of the EMBO Publications transparent editorial process initiative (see our Editorial at

<http://embomolmed.embopress.org/content/2/9/329>), EMBO Molecular Medicine will publish online a Review Process File (RPF) to accompany accepted manuscripts.

This file will be published in conjunction with your paper and will include the anonymous referee reports, your point-by-point response and all pertinent correspondence relating to the manuscript. Let us know whether you agree with the publication of the RPF.

I look forward to receiving your revised manuscript.

Yours sincerely,

Lise Roth

***** Reviewer's comments *****

Referee #2 (Comments on Novelty/Model System for Author):

Human heart samples were used that is the most appropriate type of sample/model for the study.

Referee #2 (Remarks for Author):

Thanks for addressing all the concerns and for providing a great resource for the community.

Referee #3 (Comments on Novelty/Model System for Author):

Human tissues were used for this study

Referee #3 (Remarks for Author):

The authors addressed my comments

The authors addressed the remaining editorial issues.

15th Jul 2025

Dear Prof. Lal,

Thank you for submitting your revised file. I am pleased to inform you that your manuscript is accepted for publication and is now being sent to our publisher to be included in the next available issue of EMBO Molecular Medicine.

As part of the EMBO Publications transparent editorial process initiative, EMBO Molecular Medicine will publish online a Review Process File (RPF) to accompany your manuscript. This file will include the anonymous referee reports, your point-by-point response and all pertinent correspondence relating to the manuscript. Please let us know immediately if you don't agree with the publication of the RPF.

Yours sincerely,

Lise Roth
